# University–Museum Partnerships for K-12 Engineering Learning: Understanding the Utility of a Community Co-Created Informal Education Program in a Time of Social Disruption

**Sandra Lina Rodegher** [1,*], **Lindsey C. McGowen** [2], **Micaha Dean Hughes** [2], **Sarah E. Schaible** [2], **Ayse J. Muniz** [3] **and Sarah Chobot Hokanson** [1]

1   Photonics Center, Boston University, Boston, MA 02215, USA; sch1@bu.edu
2   Department of Psychology, North Carolina State University, Raleigh, NC 27695, USA; lcmcgowe@ncsu.edu (L.C.M.); sschaib@ncsu.edu (S.E.S.)
3   College of Engineering, University of Michigan-Ann Arbor, Ann Arbor, MI 48109, USA
*   Correspondence: srodeghe@bu.edu

**Abstract:** This study explores the impact of COVID-19 on informal learning institutions, primarily science museums, through the lens of an activity kit co-created by CELL-MET—a cross-university, engineering research center—and museum partners. While formal learning organizations, like K-12 schools, play a critical role in the education process through standardized teaching, informal learning organizations also make important contributions to the engineering education ecosystem, such as by fostering engineering identity development, especially for learners and their families. This is particularly valuable for young learners from underrepresented and under-resourced communities. In this study, two questions are addressed: (1) How were museums impacted by COVID-19 and the resulting disruptions to their operations, and how did they respond? (2) To what extent were museums able to implement and adapt EEK! to reach under-served youth in the face of social disruption? When the world was experiencing social disruption from the spread of COVID-19, the authors realized they had an opportunity to test the utility and adaptability of their model of engineering activity co-creation. Approximately six months into the launch of both EEK! and the global pandemic, a 29-item survey was distributed to EEK! recipient institutions. Of the museum respondents, 97% reported experiencing full closures and 73% reported layoffs and furloughs. Despite these challenges, 78% implemented EEK!, with 70% of the institutions creating new virtual programming, and 38% adapting EEK! for remote facilitation, including real-time virtual events, self-guided activities, and at-home activity kits. Museums were equally impacted by COVID-19 policies and closures, but have not received the public attention and support that K-12 schools have received. Nonetheless, they have responded with ingenuity in using and adapting EEK!. Given their K-12 partnerships, flexibility, and ability to engage learners, museums are undervalued collaborators for universities trying to impact the K-12 engineering education ecosystem.

**Keywords:** informal education; engineering education; STEM; engagement; co-creation; diversity; equity; inclusion; K-12 learning; teaching and learning; educational partnership

## 1. Introduction

In March 2020, the start of the COVID-19 pandemic resulted in unexpected mass school closures across the world to protect the health and safety of students, educators, and the broader community. In the United States, school building restrictions forced traditional in-person instruction to shift to online learning, causing social and political impacts on the field of education that are still lingering today. More than 50 million students were physically displaced from the formal classroom for weeks to months, time frames that were largely dependent upon return-to-school policies as outlined by individual school districts. While the full extent of the ramifications caused by COVID-19 school closures may be impossible

to assess, it has been widely reported that K-12 students are still experiencing persistent academic and behavioral effects as a result of educational and social isolation [1–4].

The pandemic publicly revealed the vulnerability of the U.S. education system to social disruption. Many people now associate widespread school closures with the COVID-19 pandemic, but we emphasize that large-scale classroom disruptions have been experienced and will continue to be experienced across the country because of various social and environmental factors, such as climate change and the ongoing national teacher shortage [5,6]. A study by Wong et al. [7] found that in the U.S., during the two-year period between 2011 and 2013, more than 20,000 unplanned school closures occurred, affecting over 27 million students and 1.73 million teachers (p. 1). In the news and in education research, we typically see the effects of these closures presented as losses or deficits on the student's part—the "achievement" gap. However, by failing to address the underlying, systemic challenges that inequitably affect students—especially students of color and students from low socioeconomic backgrounds—these data reports contribute to and exacerbate social and educational inequities [8]. For example, schools that were already under-resourced prior to the pandemic now have ongoing disruptions that have been found to have long-term, negative impacts on student success, such as class sizes that are two-to-three times larger than recommended, and four-day school weeks instead of the traditional five-day school week [9].

The U.S. workforce faced similar insecurity during and following the pandemic, with some scholars noting unemployment rates on par with the Great Depression [10]. Though it is uncertain how COVID-19 will impact the U.S. workforce in the long term, innovation-based engineering jobs have held steady to address the technology and healthcare challenges associated with the pandemic. However, diversity within the science, technology, engineering, and mathematics (STEM) workforce continues to lag behind U.S. demographic trends at a time of increased demands on the STEM workforce [11]. Engineering, in particular, has struggled to attract and retain diverse populations, including women and Black, Indigenous, and People of Color (BIPOC), despite gains in other STEM fields. Research suggests—and we stress—that the barriers to an engineering career begin with the inequalities in academic exposure and domain identification in educational contexts that occur in early elementary and extend to university and industry cultures, all of which are exacerbated by a lack of diverse representation and access at every step of the way. It is inherently unjust to have a system that does not extend equal opportunity to all subsets of the population, whether intentionally or otherwise, and the unrelenting state of unequal educational opportunity has been further revealed during a time of widespread school closures and teaching and learning insecurity.

To support public educators in addressing these opportunity gaps, we highlight the potential benefits of using informal K-12 teaching and learning initiatives via inter-institutional partnerships as a shared solution to learning loss and the psychosocial threats resulting from educational isolation. By considering the broader engineering education ecosystem, we envision informal education institutions, like science museums and other out-of-school-time learning programs like robotics clubs and summer camps, as partners to formal education institutions, like public K-12 school systems, for the support of students and teachers. While many informal educational contexts, such as science museums, are equally as vulnerable as public schools to widespread closures, they have demonstrated the capacity to be extremely adaptive and resilient in the face of social disruption [12–14]. Furthermore, since the 1990s, museums have charged themselves with developing a culture of anticipating and adapting to changes in societal needs [15].

Since its initial inception, CELL-MET, a National Science Foundation-funded Engineering Research Center (ERC) that spans 10 universities with Boston University, Florida International University, and University of Michigan as the core institutions, has prioritized partnering with and supporting informal learning institutions, as critical cultural and community centers, hubs for learning, and key partners for broadening participation in engineering through engaging young learners in fun, innovative, and meaningful ways.

In what follows, we first provide context for the role of informal learning institutions in education and why they are an important societal resource, capable of attracting learners from diverse backgrounds in service of broadening participation, as well as quickly pivoting to accommodate and address societal disruptions. Then, we offer deeper insight into CELL-MET's efforts to broaden participation through the introduction of the Engineering Engagement Kit (EEK!), CELL-MET's premiere informal education program designed for children from 6 to 12 years of age and their families. We specifically focus on how the kit was collaboratively created with design elements that support adaptation by a wide range of informal learning contexts. EEK! distribution began in February of 2020, just as COVID-19 forced broad-scale closures and shifts to remote methods of teaching and learning engagement, instantly putting the program's adaptability to the test. We also present survey results documenting EEK!'s implementation and adaptation amid the pandemic, a time when the overarching operations of informal learning institutions were significantly disrupted. Through exploring how they were impacted by COVID-19 and how they used and adapted a university–museum co-created activity kit to respond to this disruption, we gain insight into how informal learning institutions could be more effectively engaged with and supported, to both broaden participation in STEM and mitigate the impacts of current and future social disruption that exacerbate existing disparities within STEM education and the engineering workforce.

### 1.1. Background: Integrated Informal Learning

The term "ecosystem" was created to characterize the type of organisms that inhabit a given space and, more importantly, the nature of their interactions and the resulting functions and outcomes [16]. By operationalizing an ecosystem lens, we actively challenge the notion that children develop—academically, emotionally, or socially—within a formal schooling silo. Children do not simply attend school to learn all there is to know about the world and then go home having sufficiently learned it. In his work on the ecology of human development, Urie Bronfenbrenner [17] suggested that much research "is carried out not in reality, but in artificial settings believed to be more conducive to scientific investigation" (p. 439). The field of educational psychology tells us that the reality of child development includes significant influence by social contexts and interpersonal relationships, both of which impact a child's worldview and ability to overcome personal challenges and disruptions experienced during a person's lifespan [18]. Bronfenbrenner's [19] theory of ecological systems models is this: visually conceptualized as concentric rings of influence, the individual at the core of the model is discursively influenced by their microsystem (such as immediate family and friends), their mesosystem (such as their neighborhood or school system), and other factors that make up the ecological exosystem, macrosystem, and chronosystem (which include the larger social, political, and cultural aspects of the individual's world) [20].

This systems approach to social ecology has clear implications on educational engagement, opportunity, and attainment for children. The model has been adapted in engineering education to describe a holistic viewpoint on the broader societal context and complex interplay of social interactions that help or hinder student engagement and learning within engineering contexts [20,21]. Intervening at the interplay between these dynamic systems, where there are significant barriers that limit the participation of underrepresented identities in engineering learning, is key to broadening participation in the field and, thus, is a key goal of EEK!.

Though the majority of our partners were science museums and EEK! was designed to fit best within that setting, we preferred not to limit the potential recipient pool. Instead, we allowed museums to self-assess EEK! utility and, ultimately, included interested children's, history, and natural history museums as recipients and partners. As such, it is important to understand the broader context in which science museums sit. They are just one example of informal learning, which also includes programs like summer camps and afterschool programs, as well as other types of museums—such as those mentioned

above. "Informal learning" can be defined in a number of ways, but is generally understood as educational experiences that occur outside of the mandates of formal curriculum or institutional structures [22]. Two key factors associated with informal learning are that (a) learning is open and learner-driven and, in turn, (b) it allows for the whole learner, including their background and interests, to be brought to the table [23].

A critique of the discourse around informal and formal learning in engineering is that the boundaries between the two areas are not as discrete in practice as they may initially appear [24]. For example, learner-driven, informal learning activities can occur within a formal education environment led by either a teacher or an informal educator via an outreach program. That same learning experience can be woven in to address curriculum requirements, further formalizing it. Similarly, an informal learning institution, such as a science museum, can develop a structured curriculum for a summer camp that uses content standards, while still focusing on processes of guided discovery learning. Though the flexibility of education muddies the delineation between boundaries, this blurred boundary is also indicative of the close, reciprocal relationship between these two learning contexts. Despite being frequently treated as very separate, it is worth noting that, historically, this makes sense; many museums originated from universities as a method for sharing their collections and educating the public Thus, as we elaborate in the sections that follow, informal learning institutions have threefold value for broadening participation in engineering, as they provide direct engagement with under-served audiences via youth and their family members who may attend programs with them; an extension of school-based learning in partnership with teachers and learners; and bi-directional professional development opportunities and other teaching and learning resources directly to educators.

### 1.1.1. Museums, Boundary Objects, and Society

Informal learning offers unique opportunities for society to engage with and explore complex topics. Science museums can help people debate critical questions, and better parse and debate topics in service of deeper learning and increased agency via exhibits, activities, and demonstrations [25]. This is in alignment with their ability to provide content, via exhibits and activities, that also function as boundary objects. "Boundary objects" are objects that are concrete enough to have shared meaning, but flexible enough that diverse perspectives can be engaged and a dialogue created [26]. Though used broadly, the process of engaging with boundary objects had its origins within the museum sector through partnerships with universities where boundary objects, such as a specimen, machine, or piece of art, were used to facilitate dialogue between a diverse group of scientists and citizen scientists. Through the use of boundary objects, informal learning institutions create an opportunity for participants to expose differences in knowledge sets and understandings. In other words, boundary objects do not force a consensus around a single understanding, but instead allow for translation, knowledge sharing, and innovation across diverse populations and perspectives. Though the literature on boundary objects focuses on the benefits to the group, perhaps the most important benefit is to the novices who may rapidly gain agency through the process of engaging with an object either through responding to the object, as in the case of art, or by directly manipulating it, as in the case of gameplay or development [27]. The engineering activities included in EEK! were created to encourage their utility as boundary objects by not having a "correct" answer, but instead leaving space for a discussion of differing perspectives and knowledge spaces. Hence, informal learning institutions can use boundary objects to engage diverse participants in ways that allow for deeper internalization in the development of an engineering identity, which often predicates the pursuit of careers in engineering and other STEM fields [28].

Beyond providing a space for engagement and identity development, museums also offer a safe space for activism, social work, and justice. It is understood that museums provide spaces for families and the broader community in ways that are linked to their missions, such as gathering and engaging together [29], and giving a venue to learn about and discuss current events and science that may be critical for their visitors. However,

they also engage in more generalized social work, such as specialized programming for individuals with serious behavioral and physical health concerns [30], and even provide a refuge from inclement weather [31]. Science museums are wholeheartedly owning their roles as spaces for activism and social justice, whether mitigating sustainability issues, supporting refugee communities, or addressing intolerance [32,33]. Because of this focus on activism, social work, and justice, museums are also well positioned to be effective partners in efforts to broaden participation in engineering. Likewise, if museums and other informal learning institutions have the flexibility and capacity to provide support to other societal challenges, it stands to reason that they may be equally poised to support society during times of social disruption.

### 1.1.2. The Relationship between K-12 Informal and Formal Learning

In addition to providing direct benefits to families and communities, informal learning institutions can also strengthen the impact and reach of formal learning. While formal approaches to education provide a necessary standardization across a larger percentage of K-12 learners, the need for schools to standardize content constrains teachers' ability to offer open-ended discovery that is engaging for learners with diverse interests [34,35]. Public K-12 schools must use their time to teach state or federally determined standards and then prepare students for, and proctor, mandated testing. In one U.S.-based survey, testing alone took as much as 20 to 50 h per year, and preparation took up to an additional 110 h [36]. Completing these tasks efficiently can be particularly challenging in under-resourced schools, which tend to have higher student-to-teacher ratios, larger teacher turnover rates, and higher rates of alternative teacher certification [37]. The impact of formal education is further limited because school-aged youth spend just 13% of their waking hours in school [38]. In the wake of these challenges, the use of informal learning institutions to supplement formal instruction is well documented in the STEM education literature, and science museums and K-12 schools have a history of partnering in a diversity of ways [39,40]. The open-ended, exploratory nature of informal learning institutions has historically complemented the instructor-driven, standardized content of schools. Further, visiting science museums has been an effective means for enhancing student learning [41]. Especially when it is considered as a part of the broader education ecosystem, informal education is not a threat to formal education—education researchers, like Dorie et al. [42], have emphasized that the "the presence of informal learning environments does not override the use of formal learning environments", but rather that the two environments should complement each other so that students can gain from one what they cannot gain in the other. Indeed, partnerships between schools and informal learning institutions, like science museums, can support the development of both students and teachers.

### 1.1.3. Informal Learning Institutions as Resources for Educators

Science museums are known as being sites for school visits, but what is lesser known is *how* they engage with teachers. They may partner with a teacher in advance of a school visit to refine pedagogical content used during student visits [43]. Some informal science learning environments receive a large amount of teachers' time, with 35% of informal learning organizations reporting 25 h or more of contact time with teachers, which is primarily spent conducting professional development (PD) [44]. Museum educators can provide teachers with inquiry-based PD opportunities and curriculum to be used in the classroom, which furthers teachers' self-efficacy in project-based instruction, a teaching strategy that is clearly in line with design-based engineering education standards [45,46]. The ability of science museums to provide in-service science and mathematics teachers with extended PD is vitally important for the inclusion of engineering education in K-12 schools in the U.S. Despite the inclusion of engineering concepts in the widely adopted Next Generation Science Standard (NGSS), most K-12 teacher education programs still do not include engineering education content or courses, which leads to low teacher confidence in including engineering concepts in their instructional practice [47,48]. Because of their

ability to deliver quality, adaptable PD, science museums are an underutilized mechanism for addressing these issues in teacher education. The connection between schools and museums can serve to increase educational access and the strengthening of student STEM identity, if connections between formal and informal learning are built and maintained [49]. As discussed earlier, social disruption will continue to be a pressing issue for schools, as the world reacts to increased educational challenges associated with climate change, teacher shortages, and in-school violence, such as school shootings (among other forms of social disruption impacting the education ecosystem); these relationships will gain value and importance. As such, this study presents an area from which formal education could glean ideas, inspiration, and insights from informal educators, as well as find mutually beneficial support and resources, so that both types of institutions can more effectively engage and educate young learners.

*1.2. Research Context: CELL-MET*

Nanosystems Engineering Research Center for Directed Multiscale Assembly of Cellular Metamaterials with Nanoscale Precision, or CELL-MET, is taking a robust approach to broadening participation in engineering. CELL-MET, which is housed at Boston University but spans 9 others, is focused on developing a synthetic heart tissue that can be used to repair the cellular damage caused by heart disease. Because developing cutting-edge technologies to address this critical challenge for global health, which has a disparate impacts on BIPOC populations, requires the inclusion of diverse perspectives, broadening participation in the engineering workforce is an equally critical component of CELL-MET's mission. CELL-MET has developed a variety of formal and informal learning opportunities to support pre-college engineering education and engagement.

These programs are designed to build engineering identity and self-efficacy, as well as to attract diverse K-12 learners toward careers in engineering. CELL-MET's formal efforts include teacher trainings, curriculum development, and research experiences to support teachers in developing their own curricular shifts. CELL-MET's informal education programs use university–museum partnerships to broaden participation by strategically integrating research into informal learning environments through programs that complement and reinforce the learning that happens within formal classrooms. In particular, CELL-MET has integrated engineering and heart-health topics into the development of EEK!, creating an adaptable boundary object that is intended to accommodate a diversity of settings, learners, and institutional dynamics and relationships.

1.2.1. Theoretical Foundations Underpinning the EEK! Development and Implementation Process

In this section, we describe the theoretical perspectives that underpin the collaborative co-creation process that guided EEK! development, implementation, and ongoing adaptation (see Section 1.2.2 for addition details on the development process); participatory design; the collaborative chain-link model of innovation; and diffusion of innovation. Participatory design is intended to increase community knowledge and capacity, while also making the product being created more useful to target users [50,51]. Co-creation in educational settings has been identified as critical for creating inclusive learning environments and tools that embrace diversity [52] and may increase the duration and positive affect associated with the co-creation process and outcomes [53].

The collaborative chain-link model of innovation similarly relies on stakeholder participation [54]; however, it offers a different framing. From this perspective, the EEK! program can be viewed as a social innovation: a new program or approach "designed for the solution of particular social problems" [55] (p. 422). The chain-link model of innovation views the innovation process as a complex one, with each step informing the next and feeding back through the cycle for further refinement. In this model, stakeholder involvement facilitates buy-in, allows for access to a broader network of support, and can enhance program development because the knowledge and skills of both the user community and researchers

are available throughout the process [56,57]. The literature on diffusion of innovation has also demonstrated that novel programs are most readily adopted when they fit with the organization's mission, resources, and existing programming [58]. Further, the process of adapting innovative programs to the local implementation context leads to a greater sense of program ownership and buy-in, which can contribute to overall program effectiveness and long-term utilization [59]. Blakely et al. [60] found that programs that were adaptable to the local context while maintaining fidelity to core program components were more effective than programs that were not adaptable. These authors also found that co-creation, by program designers and organizations adopting the program, allowed designers to provide technical support throughout the implementation process. This technical support helped adopting organizations strike the right balance between fidelity and adaptation. Here, we use the definition of implementation fidelity offered by Mayer and Davidson [55]: "the degree to which an innovation is implemented in a manner similar to the original demonstration model" (p. 429). Conversely, they define reinvention as the process "of interaction between the innovation and organization that shape adaptation" [55] (p. 429). These authors have reported on the results of three large-scale studies that examined the dissemination of seven education and criminal justice programs across 70 organizations. The programs that were rated as (a) high-fidelity implementations, or those that carried out two specific types of changes: (b) "modified reinventions", in which the adopting organization made changes to the program model and (c) "addition reinventions", in which the adopting organization made additions to the original program model. The results indicated that modifications to innovative programs were not associated with changes to program effectiveness, but that both high fidelity and addition reinvention were positively correlated with program effectiveness.

While co-creation is not novel, COVID-19 provided a unique opportunity to gain a deeper understanding of co-creation and its impacts across a diverse range of informal learning institutions during a period of social disruption. Since the kits were co-created with CELL-MET researchers and museum partners, this participatory, chain-link process of innovation, refinement, and adaptation has continued seamlessly throughout the implementation process. Indeed, the close partnership between the kit designers and adopting museums was another critical mechanism that allowed for swift program adaptation in the face of unprecedented organizational and operational changes occurring at all levels of the education ecosystem (formal and informal) in response to the COVID-19 pandemic.

### 1.2.2. EEK! Content

CELL-MET's informal education efforts hinged on the creation of EEK!, which was designed to engage 6- to 12-year-old youths and their families in engineering and heart-healthy activities. Though exploring the nuance and full development of EEK! is beyond the scope of this paper, there must be some information provided to contextualize the current study. In what follows, we briefly summarize (a) how co-creation was employed throughout the design and implementation of EEK! and (b) what EEK! contained, along with a few small examples of adaptions.

There were three key groups of contributors to the creation of EEK!. First, museum partners contributed their expertise in activity creation and facilitation and their insights into the broader needs and trends in the museum's world and their local community. Second, CELL-MET researchers provided their scientific knowledge toward the development of the heart-related activities and broader engineering knowledge, alongside insights gleaned from their personal engineering education experience. Finally, the EEK! design team, who had previous experience partnering with hundreds of museums in over 30 countries on various projects, sought to function as a boundary spanner and facilitator of both the process and products, while also providing necessary consistency and capacity to execute. This "capacity to execute" was important to lessen the burden on the co-creation contributors, allowing more researchers and museum partners to participate by allowing for lower levels of time commitment. In short, the design team's support allowed the

researchers and museum partners to participate in critical decision making and content development, without placing an undue burden on their pre-existing responsibilities.

We have broken the co-creation process into five stages and highlight seven key co-creation inputs or inflection points. Before articulating these stages and events, we note that co-creation is an ongoing process and dialogue and that there are many smaller, informal instances of co-creation not captured as "inputs" (e.g., impromptu brainstorms with researchers about the best metaphor for describing polymers, or an unexpected phone call from a museum partner because they had an idea for a heart-related activity and wanted to share). This exclusion is a reflection of the scale of the events rather than a judgment of their importance.

Co-creation efforts began before EEK! development in the project definition phase, with preliminary conversations with science museums' leadership, explainers, and content development professionals playing a key role in defining the scope and focus of the, at that point undefined, engineering education initiative. Specifically, this was how the development of an engineering kit, over other types of museum resources, was decided upon. Through several conversations, the EEK! design team asked museum professionals (a) whether engineering education content would be valuable and, if so, (b) what would be the most useful form in which to receive this content.

During the ideation phase of development, six science museum professionals from different institutions, and professional designers and illustrators who focus on youth audiences were brought into the project. They were selected based on (a) their expertise, (b) their work serving historically excluded communities, and (c) their diverse experiences as individuals. Together with five CELL-MET researchers and two EEK! design team members, they brainstormed content, learning objectives, and design guiding principles. The guiding principles were created to support ongoing adaptation for fit at the institution, educator, and learner level.

During the initial creation and testing phase of development, both direct and indirect feedback was gathered from 69 informal education professionals from 16 museums (including two from the ideation process) and over one thousand young learners and their families who participated in piloting the activities as part of the co-creation process. In the early stages, this looked like presenting rough ideas to small groups of informal educators and in one-on-ones with children and families. In the later stages, it included larger-scale pilot testing that relied on a combination of observation, unstructured interviews, and focus groups to gather feedback and suggestions. We continued to use co-creation in the implementation phase of the project by intentionally designing the activities to be adaptable to various local contexts as well as diverse learner needs and interests, a significant consideration that any educator, whether formal or informal, must address.

To identify EEK! recipients during the distribution process, museums began with the EEK! application process. A key criterion for museum selection was their self-reported capacity for broadening participation via diversity of audiences served and types of programs offered (more detail on broadening participation efforts can be found in Section 3.3). Two museums that participated in the design process received EEK! alongside 49 new museum partners. The 51 institutions varied in their specialties, preferred facilitation methods, the communities in which they sat, programmatic offerings, partnerships, and regional and institutional structures with which they interacted. The physicality of the institutions also varied, with some museums being entirely mobile, particularly in rural areas where it was easier to visit remote communities rather than having a physical space for the community to come to. Including flexibility as a design component also proved critical for rapidly adapting the kits in response to the COVID-19 pandemic. Educator capacity-focused training prioritized ongoing co-creation by focusing most of the two-day training sessions on informal educators working with CELL-MET researchers to brainstorm and share potential modifications and extensions, based on the characteristics and needs of their unique institutions and the communities they served. Ongoing co-creation after distribution took place both formally, through group meetings, and informally, through

one-on-one discussions and support based on the participants' unique needs. Thirty-seven survey respondents reported reaching 13,554 learners and their families. This number is conservative, with several participants sharing via personal communication that they did not report on reach via activity downloads, video views, or—in the case of one science center—the EEK! news story reach. See Figure 1 for a visual overview of the EEK! design process and key inputs.

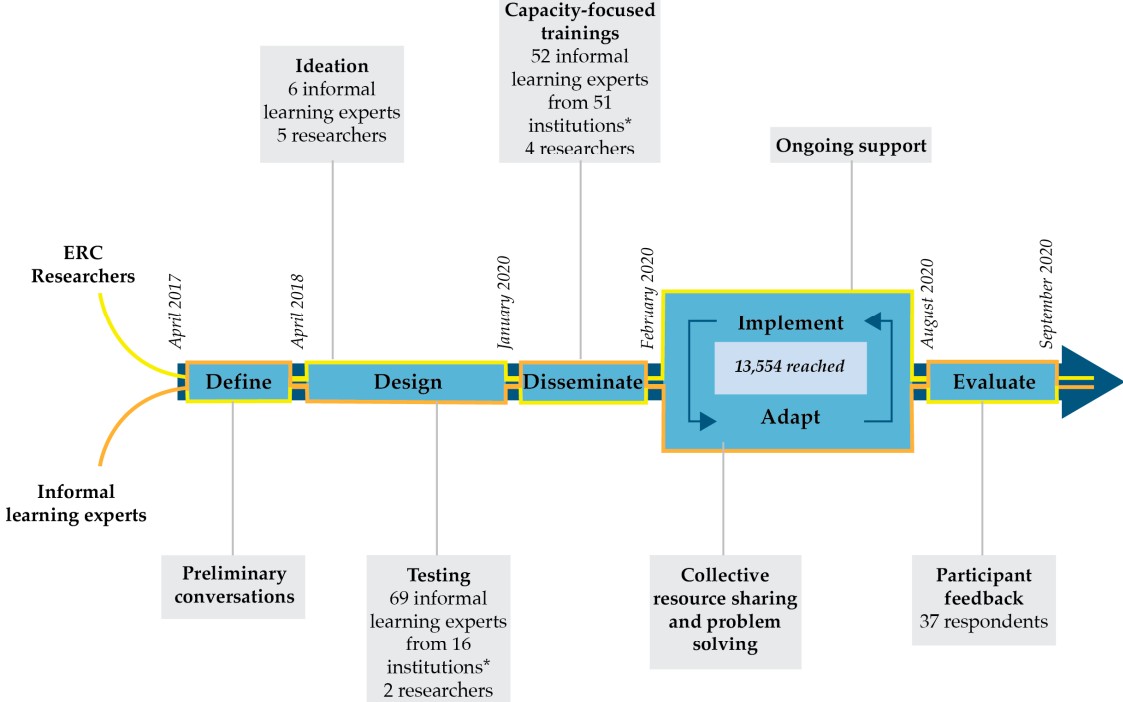

**Figure 1.** EEK!'s community co-creation driven development and distribution process.

As a result of the preliminary ideation session, EEK! content was designed around guided discovery learning theory, which consists of allowing the learner to engage their whole self in a process of self-guided exploration, past experience, creativity, and safe failure [61,62]. Interestingly, the characteristics of discovery learning pair nicely with the concept of boundary objects, which are similarly designed to allow for and encourage diverse perspectives to engage in dialogue and reflection. In addition to having high efficacy [63], discovery learning is well aligned with design-based learning and the engineering habits of mind, both of which also encourage the use of creativity and exploration in problem solving [64,65]. Further, the free-choice learning that the science museum setting encourages is effective at building science identities because learning is directed by personal interests and curiosity [66]. Ultimately, the one component that science museum partners tend to share is a focus on manipulable, physical objects, and this was explicitly built into the "hands-on, minds-on" component of the kit.

In total, the kit contains seven flexible, "hands-on, minds-on" [67,68] activities, and instructions for an additional eighth take-home or extension activity. The activities range from card-based activities to full-body games. See Figure 2 for brief descriptions of the eight activities. While some of the activities focus more heavily on engineering concepts, like Cell Posts, which asks learners to use the engineering design process to simulate "building heart tissue", other activities, such as You are an Engineer, focus less on engineering skills and more on engineering identity and interest in service of lessening cognitive barriers and increasing intrinsic motivation. Though not the focus of this paper, the authors posit that increasing intrinsic motivation is critical to the field of engineering education, as it increases resilience in the face of challenges.

| | | | |
|---|---|---|---|
| In **LAB COLLAB** players are members of a lab, racing the clock to build heart tissue. They quickly roll dice and flip cards to reveal "new research findings," all while exploring the benefits - and challenges - of collaboration. | **CELL POSTS** engages learners in the engineering design process through CELL-MET's real-life challenge - growing heart tissue. Participants must attach their "heart tissue" to the post structure and "mature" it without damaging it. | **FEEL THE BEAT** explores heart behavior and health. Learners use a device created by engineers, a pulse oximeter, to observe their heart rate and how it changes with different activities. | **HEART BEATS** is a full-body game introducing heart knowledge and teamwork. Participants must work together to model behaviors of the heart and the circulatory system. |
| **REAL OR IMAGINED** highlights the importance of creativity in engineering. Players are presented with card featuring fantastical "inventions" and they must guess which are real. | **YOU ARE AN ENGINEER** uses cards to illustrate the many types of engineering through familiar activities. Learners then graph their results to see what type of engineer they are most like. | **TROUBLE IN THE TOY FACTORY** introduces product testing in engineering. Learners assess the performance of two balls that appear the same but function surprisingly differently. | **BOUNCY BALL CHALLENGE** engages learners in make their own polymers. Participants are given a formula and then modify it to create the bounciest ball possible. |

**Figure 2.** Summaries of EEK! activities.

Due to differing institutional and learner needs, the activities were designed to be flexible in terms of duration. Each engineering activity was scaffolded so that it could be completed in as little as five minutes or expanded to take an hour or longer. This was done to accommodate both the needs of the institutions and the learners. Returning to the example of You are an Engineer (see Figure 3), participants look at the front of a series of cards and select all the activities that align with their interests. They then flip the cards over to reveal information about the associated engineering field, and tally their "score" for each type of engineer listed at the bottom of the card to see which engineer they are most like. This activity is flexible in two ways. First, the cards' written content was co-developed with graphic designers to create visual hierarchy that allows users and facilitators to easily filter in and out components based on learner ability. For example, a new reader or young learner might focus heavily on element A with some exploration of element B, whereas a more advanced learner might engage in a more robust discussion that includes debating the categorization used in D or brainstorming the question posed in C. Second, activity extensions include turning the questions posed in C into full design challenges and links to pre-existing design challenges and laboratory experiments aligned with each type of engineer listed on the cards. In this way, these activities have been designed to be adapted for all ages and the full gamut of the science museum experience.

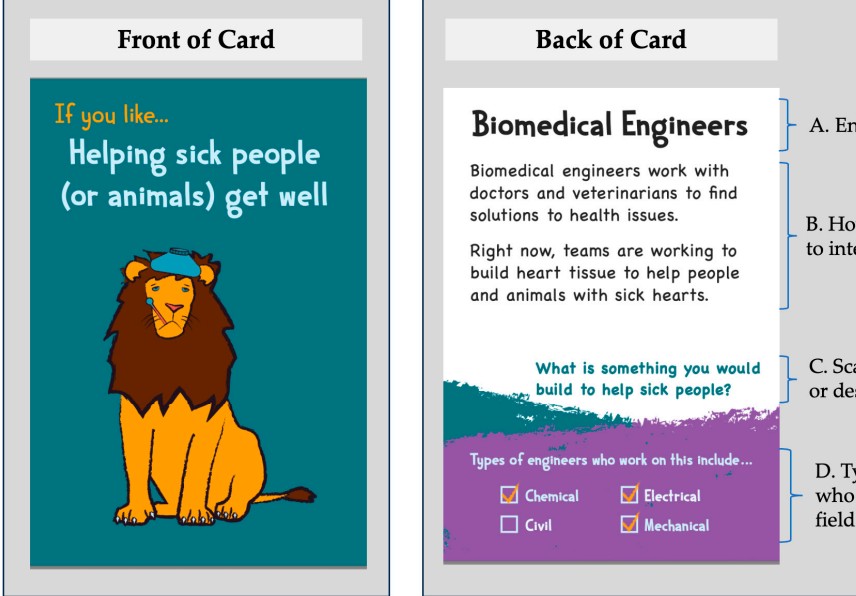

**Figure 3.** Sample You are an Engineer card.

### 1.3. The Current Study: COVID-19 Context

Recent research and societal focus have been aimed at (a) the responsiveness of the formal education system to COVID-19 and (b) the lingering effects of related school closures and disruptions on K-12 learners. This is for good reason—schools provide safe spaces, meals, academic teaching and learning, and social interactions for students [69]. However, schools cannot effectively operate in isolation, and informal learning institutions are an important component of the education ecosystem, particularly for social/emotional learning and engineering identity development. Though they have great potential for supporting the engagement of diverse and under-resourced populations, museums are often underutilized [49]. Given that the global pandemic has increased inequality for under-resourced and under-served communities and widened the opportunity gap, we posit that there is great importance and value in reflecting on (a) the impacts of COVID-19 on museums and, more importantly, (b) how carefully scaffolded partnerships can create additional resilience in the face of social disruption.

Thus far, despite museums having enormous social value, there has been significantly less attention given to the impacts of COVID-19 on these institutions. Indeed, in a comparable Google search conducted at the time this article was written, "impact of COVID-19 on museums" (199 million hits) received just 4.2% of the hits that "impact of COVID-19 on schools" received (4.74 trillion hits). Drilling down further, the "impacts of COVID-19 on science museums" had far fewer, with just 285,000 hits. Though it is to be expected that museums, and science museums in particular, would receive less attention because student attendance at these institutions is not required, we must not overlook the support and engagement they offer to families and their role in the education ecosystem. Science museums have observed the needs of essential workers and low-income communities along with the gaps created by school closures and adapted by offering childcare and learning pods, partnering with farmers and restaurants to provide free food to supplement the meals lost through school closures, developing virtual fieldtrips and workshops to provide breaks to burnt-out teachers, and countering COVID-19 misinformation through their content [70]. Indeed, it is possible that families are relying *more* on museums as a result of pandemic-related disruptions.

Three EEK! trainings were completed in January of 2020 and the kits were distributed in early February. This gave many museums just a few weeks to use EEK! before they began reporting layoffs and closures in March. Rather than put the kit on hold indefinitely, the EEK! development team worked with the adopting museum partners to rapidly adapt it in three key ways. First, given the unprecedented nature of COVID-19 and the variation in state responses across the nation, we convened community check-ins, which began as bi-monthly meetings for any kit recipient organizations' staff members, as well as other informal education professionals who expressed interest. These were reduced to monthly check-ins in July. We adopted this active support model of innovation dissemination because it was a natural continuation of the collaborative chain-link process used to design the kits, and because an active technical assistance or consultation approach has been shown to lead to greater implementation success [58,71]. Second, since many of the participants had scheduling conflicts with the community check-in, we produced a monthly newsletter. This active support feature also allowed the participants to disseminate information about the EEK! program more broadly within their home organizations and to their partners. Finally, operational modifications were made to EEK! structures and layouts to better accommodate socially distanced informal education, which included virtual video-based learning, as well as physical take-home kits that the museums were creating. Some of these modifications were small shifts (see Figure 4 for sample shifts); however, they created pathways for use that would not have otherwise existed. In contrast, other modifications were more robust, such as the creation of a heartrate-focused scavenger hunt or co-created modifications to support use in in-patient mental health facilities, where many materials such as pens and paperclips are prohibited.

| | Original | Modification 1 | Modification 2 |
|---|---|---|---|
| **Appearance** |  |  |  |
| **Medium** | 5×7 inch cards, printed on synthetic paper | 10×7 inch presentation to be viewed from a computer | 8.5×11 inch cardstock |
| **Facilitation** | Facilitator guided | Facilitator or self-guided | Self-guided |
| **Timing** | Synchronous | Synchronous or asynchronous | Asynchronous |
| **Venue** | In-person | In-person or virtual | In-person or remote |

**Figure 4.** Sample EEK! activity and two modifications initiated by partner institutions.

The purpose of this study is to understand how an informal engineering education intervention—the EEK! kits—was implemented under conditions of social disruption within museum settings. In particular, we explore the ways in which the kits were adapted in response to the changing public health and organizational conditions under which the kits were implemented, and the role of university–museum partnerships in supporting those processes. Though the kit is a very specific type of engineering education intervention and the COVID-19 pandemic was a unique form of social disruption, we use them as a proxy for any engineering-focused informal education resource that may be co-created in partnership with informal educational institutions as they navigate social disruptions. Specifically, we seek to address the following research questions:

**RQ1.** *How were museums impacted by COVID-19 and the resulting disruptions to their operations, and how did they respond?*

**RQ2.** *To what extent were museums able to (a) implement and (b) adapt EEK! to (c) reach under-served youth in the face of social disruption?*

Understanding how these museums have adapted may allow for the creation of new initiatives, supports, and programs to bolster and enhance the resilience and adaptive capacity of informal learning institutions, and the larger education ecosystem more broadly.

## 2. Materials and Methods

The current study uses an exploratory and descriptive research design, combining qualitative and quantitative methods, to examine the impact of social disruptions on museums and the role of partnership-based, co-created programs in their efforts to adapt to those disruptions, using the EEK! program as a case study. EEK! was developed, implemented, adapted, and evaluated over a 2.5-year period, with program development beginning in 2018. The program was officially launched in early 2020 and evaluation data were collected in Fall 2020 (see Figure 1). In what follows, we focus on the 37/51 institutions that received EEK!, responded to the survey, and consented to share their experiences during the first six months of a global pandemic. A 29-item survey was distributed to key museum staff at the 51 EEK! partner institutions as part of EEK! program evaluation. It is important to note that the decision to survey museum staff rather than EEK! participants

was informed by an understanding of important considerations and challenges associated with evaluations of informal engineering education [72,73] (Fu et al., 2019; Teasdale, 2022). In particular, there are concerns about ecological validity, as many traditional education evaluation methods violate the very principles that make informal education valuable. For example, pre–post tests, interviews, and formal surveys administered to students are considered to be disruptive to the engaged nature of informal learning contexts [20].

The survey, developed in partnership with an external evaluator, included rationally created items based on the program content and study goals. The survey included a combination of forced-choice (categorical and Likert-scale) and open-ended questions focused on the impact of the COVID-19 pandemic on museum operations, organizational engagement with under-served populations, kit implementation and adaptation, and resource utilization. Examples of forced-choice and open-ended questions for each of these areas are presented in Table 1. Qualitative items were strategically included to gather details and context for the results reported in response to forced-choice items. Since the data were collected several months into EEK! implementation, questions were structured to gather data about the respondents' current and retrospective status and operations.

**Table 1.** Examples of forced-choice and open-ended survey items.

| Focus Areas | Example Items |
|---|---|
| Impact of COVID-19 on museum operations | **Forced Choice:** Please indicate the current operational status of your organization.<br>- My organization is open to in person visitors with no restrictions on building capacity<br>- My organization is open to in person visitors, but operating under reduced-capacity limitations<br>- My organization is not open to in person visitors, but is hosting virtual activities and programs<br>- My organization is not currently operating, but has plans to re-open in the future<br>- My organization is no longer in business<br>- Other, please explain<br><br>**Open Ended:** NA |
| Organizational engagement with under-served populations | **Forced Choice:** With which of the following types of organizations does your organization partner, now or within the last year? Please mark all that apply.<br>- Preschools, Elementary Schools, Middle Schools, High Schools, Community Colleges, Colleges or Universities, Organizations providing before or after school childcare, Other museums, Corporations or other industry partners, Non-profits or foundations, Hospitals/healthcare, Public health organizations, Camps, Organizations serving low income communities, Organizations serving communities of color, Organizations serving women and girls, Organizations serving people living with disabilities, Organizations serving immigrants, Organizations serving veterans, Other types of organizations (please describe), No partner organizations<br><br>**Open Ended:** Please describe what your organization does to reach underserved audiences for the EEK! program. |
| EEK! Implementation | **Forced Choice:** Please indicate which of the following EEK! implementation methods your organization is currently using. Please mark all that apply.<br>- My organization is currently implementing EEK! activities in person<br>- My organization is currently implementing EEK! activities as a live/real-time virtual event<br>- My organization is currently providing EEK! activities as a recorded/self-guided virtual activity or resource<br>- My organization provided EEK! activities as take-home activity boxes<br>- My organization is currently implementing EEK! activities in some other way. Please describe<br>- My organization is not currently implementing EEK! activities<br><br>**Open Ended:** Please explain why your organization did not implement any of the EEK! activities. |
| EEK! Adaptation | **Forced Choice:** Did you, or other facilitators, modify EEK! activities?<br>- Yes<br>- No<br>- Not sure<br><br>**Open Ended:** How did you modify EEK! activities? Why? |
| Resource utilization | **Forced Choice:** [For each resource utilized] Please indicate how helpful were each of the EEK! trainings or resources you used.<br>- Not at all helpful<br>- Slightly helpful<br>- Moderately helpful<br>- Very helpful<br>- Extremely helpful<br><br>**Open Ended:** Please comment on how the trainings or resources you used helped support your use of EEK! |

Participating museums were notified about the survey through monthly check-in calls. They then received an email further informing them about the purpose of this study, inviting them to participate, indicating that responses were voluntary and confidential (reported only in aggregate), and directing them to the survey link. Participants received up to two survey reminder emails encouraging them to respond to the survey and were re-contacted after completing the survey to obtain permission to use their completed evaluation survey responses for research purposes (response rate = 37/51, 73%). The data were analyzed to compute descriptive statistics. A qualitative content analysis was used to code open-ended comments for relevant themes. We used a grounded theory approach to identify themes that emerged from the qualitative data [74]. To address the trustworthiness of data, two coders independently developed and defined codes and applied them to comments. Codes were compared and discussed to arrive at consensus codes for all comments [75].

Of the 37 museums that responded to the survey, 70% of them identified as science museums—representing a large majority of total respondents. The remainder of the sample comprised children's museums (19%), history museums (5%), and natural history museums (5%; see Figure 5). As such, this study largely provides insight into the state of science museums and the impact of COVID-19 on their operations, as well as their ability to implement and adapt EEK! to their local contexts. Given the diversity of COVID-19 restrictions implemented at the state level and their impacts on participant organizations, it is worth noting that respondents spanned a diversity of geographic locations and contexts, with 30 states represented in the survey data (see Figure 6 for a map of respondent location).

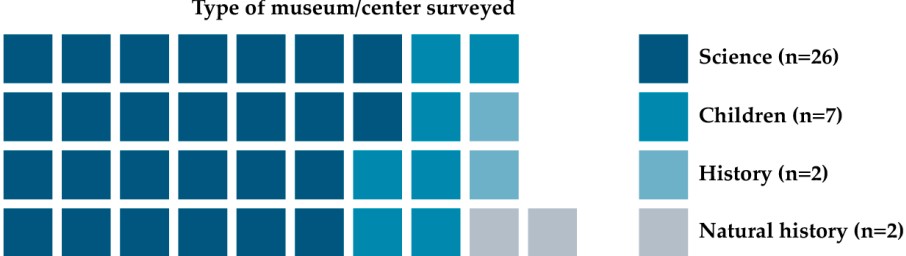

**Figure 5.** Breakdown of survey participants by institution type.

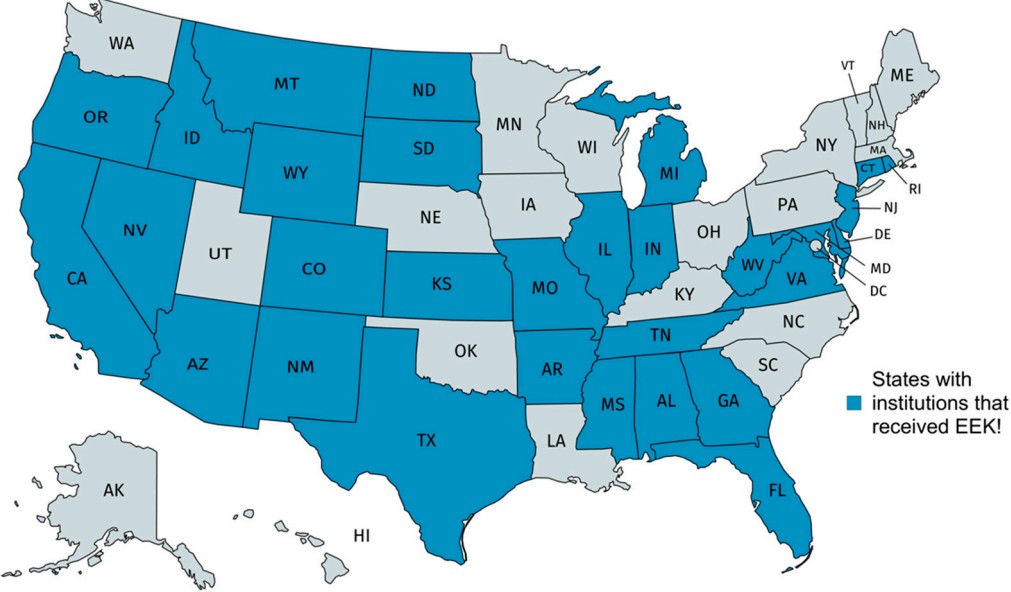

**Figure 6.** Map of institution location of survey respondents.

## 3. Results

### 3.1. RQ1: Impact of COVID-19 on Participant Institutions

In order to address RQ1, survey respondents were asked about the ways in which their organization had been impacted by the pandemic. At the time of data collection, 57% of respondents indicated their institution was open to in-person visitors but operating at a reduced building capacity. However, 32% of respondents indicated that their institution was not currently open to in-person visitors, but had adapted by hosting virtual activities and programs. The remaining 11% of respondents reported that their organization was not currently operating due to COVID-19, but had plans to re-open in the future (see Figure 7). It is important to note that none of the survey participants indicated unrestricted operations, showing that none of the involved museums remained untouched by the impacts of COVID-19. Alternatively, no respondents indicated that their organization was no longer in business, which is reassuring when many organizations had to permanently shut their doors during the pandemic. It is worth noting that similar results were found by the American Alliance of Museums (AAM), who conducted a real-time analysis of the impact of COVID-19 on museums during the pandemic. Specifically, they found that 98% of museums closed at some point in 2020, and 29% were closed as of late October 2020 [76].

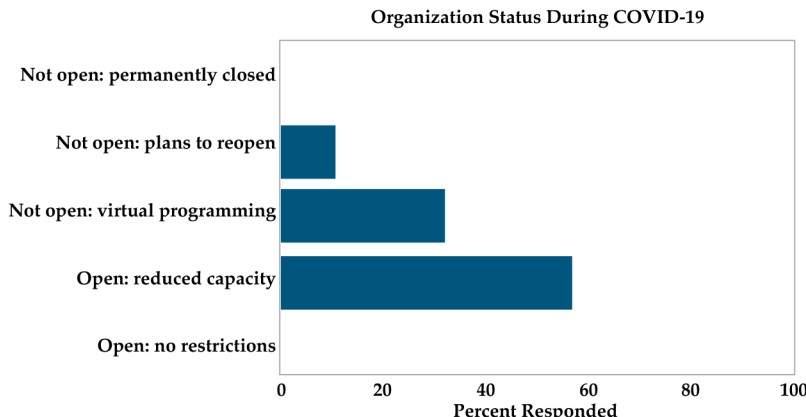

**Figure 7.** Operating status: impact of COVID-19 on museums (select all that applied during the first six months of the EEK! program).

With all respondents indicating changes in operations due to COVID-19, it is imperative to acknowledge what this meant for the organizations' staffs. AAM's survey [76] found that 53% of museums reported furloughs or layoffs; however, our sample experienced higher rates of staff reduction. A majority of the responding organizations (57%) reported that layoffs occurred in light of the operation changes, with only 16% of respondents indicating that operation changes had had no impact on staffing (see Figure 8). Moreover, 16% of museums reported furloughs, with 11% reporting only some employees currently back working, and 5% reporting all employees back working. This means that 73% of the museums involved in our study were working with a reduced staff at some point compared to pre-COVID-19 times; meanwhile, job demands are arguably higher with the large need to make significant changes and adjustments to operations in light of the COVID-19 pandemic.

Unsurprisingly, our findings reveal that not one museum was protected from the need to adapt during the COVID-19 crisis (see Figure 9). Despite a majority (70%) of organizations allowing in-person visitors at a reduced capacity at the time of data collection, all but one organization reported being closed to in-person visitors at some point due to COVID-19. With the widespread closure of museums to in-person visitors, 70% of museums reported that they converted existing in-person programming to be delivered via a virtual platform. Notably, museums not only offered existing programming virtually, but many

created new virtual events, activities, and resources, whether they were recorded and self-guided (70.%) or real-time, live programs (62%, see Figure 10).

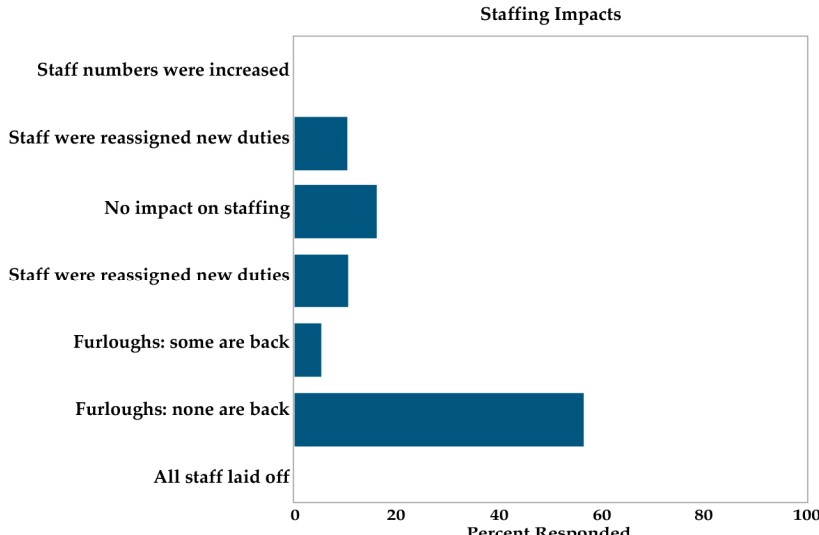

**Figure 8.** Staffing: impact of COVID-19 on museums (select all that applied during the first six months of the EEK! program).

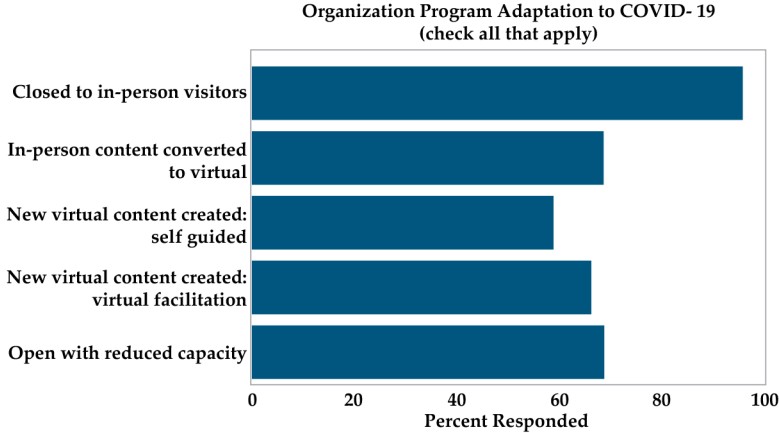

**Figure 9.** Program delivery: impact of COVID-19 on museums (select all that applied during the first six months of the EEK! program).

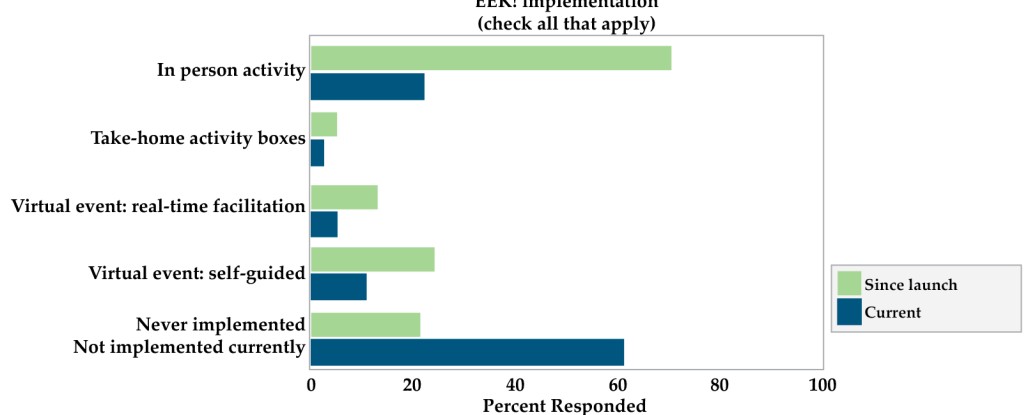

**Figure 10.** EEK! implementation: modes of delivery used currently and at any time since program launch.

*3.2. RQ2a: EEK! Implementation*

The impacts of COVID-19 on museum operations and their resulting adaptations also impacted the implementation of EEK!. Recall that EEK! kits were distributed in February, leaving little time for their implementation prior to the onset of COVID-19. This resulted in 22% of the organizations reporting having never implementing the kits. In response to an open-ended question, all of the organizations that reported never implementing EEK! cited COVID-19-related reasons, including staffing limitations, site closures, lack of access to the kits due to remote working, and canceled events hampering their ability to deliver the kits (see Table 2).

**Table 2.** Factors preventing EEK! implementation.

| Theme | % of Comments Reflecting Theme | Illustrative Comment |
|---|---|---|
| Canceled events | 63% | *"we were going to roll out the EEK! activities at Space Day, which was supposed to happen March 14th but that was canceled when the schools suddenly closed). Butterfly weeks were canceled. Field trips are all canceled and we aren't allowed to go to the schools either. It briefly looked like we were going to run camps but we ended up only having one week of a few low-enrolled (most had 2–6 kids) camps and then that was also canceled"*. |
| Site closures | 63% | *"Because of the pandemic our museum was shut down to in-person programming (and continues to be)"*. |
| Staffing | 50% | *"The staff member originally involved was laid off—in fact, 80% of staff was laid off"*. |
| Remote work | 13% | *"the EEK! kits are sitting in 3 different offices of which the staff are still working remotely with limited office access"*. |

Despite these disruptions from COVID-19 restrictions, the vast majority of respondents, 78%, reported implementing EEK! at some point (see Figure 10). Nevertheless, more than half of the respondents (62%) reported they were not currently implementing the kits at the time of data collection. This aligns with AAM's [76] finding that 67% of the survey respondents were forced to reduce their educational offerings, programming, and other public offerings due to limitations related to reduced budgets and staff capacity. This demonstrates that critical informal learning activities were restricted, hampered, and deprioritized during COVID-19—a problem that should not go unrecognized.

*3.3. RQ2b: EEK! Accommodations and Adaptations*

Many museums' efforts towards finding new creative ways to offer programming were directed at EEK!, in co-creation with the EEK! design team. Despite most of the respondents reporting that the kits fit very or extremely well (on a 5-point Likert scale: 1 = not well at all, 5 = extremely well) within their organization's budget, existing programming, and overall mission (see Figure 11), adaptations were encouraged given the diversity of needs and environments of the adopting organizations and, especially so, given the ever-changing circumstances of a global pandemic. To further understand program implementation and adaptation, we asked respondents about changes they made to the kits, as well as their utilization of the resources available through the university–museum partnership to support program adaptation.

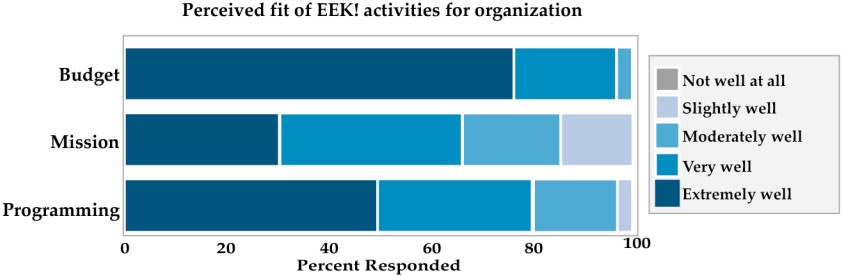

**Figure 11.** EEK!'s fit with museum learning contexts: alignment with organizational budget, mission, and programming.

The respondents were asked whether they had changed any of the EEK! activities. For the 43% who reported they had made changes, they were asked to describe how and why they changed the activities. In order to better understand what types of adaptations these represented, the responses were coded into one of the three implementation and adaptation types identified by Mayer and Davidson [55]: high fidelity (no changes reported), modifications (COVID-19 specific or other changes), and additions to the program. Over half (57%) reported no changes, reflecting high-fidelity implementation. Of the museums that reported making changes to the kits, 36% reported additions, particularly additions that integrated the kits into their larger programming to leverage existing organizational resources or make the kits more experiential. As one respondent explained:

> "Mostly added on to the materials—we allowed full use of all our maker space materials for Cell Posts, I enlarged the "draw yourself as an engineer" to a full sheet of paper for You are an Engineer, we used the happy and sad balls in combination with our Keva Planks to build rolling tracks for the Trouble in the Toy Factory activity, etc. No notable modifications to the activities themselves, just small details about how they were implemented".

Of those reporting changes, 68% reported making modifications to the kits—23% were changes made due to restrictions implemented in response to the COVID-19 pandemic, and 46% were other modifications to the kit implementation (see Figure 12). The COVID-19-specific response adaptations were primarily made around the mode of delivery–transitioning of the kits for virtual use, or to be a take-home activity. This included adapting the activity materials into more simplified worksheets and shifting more of the information and activities into "learn more" formats to be tackled independently. For example:

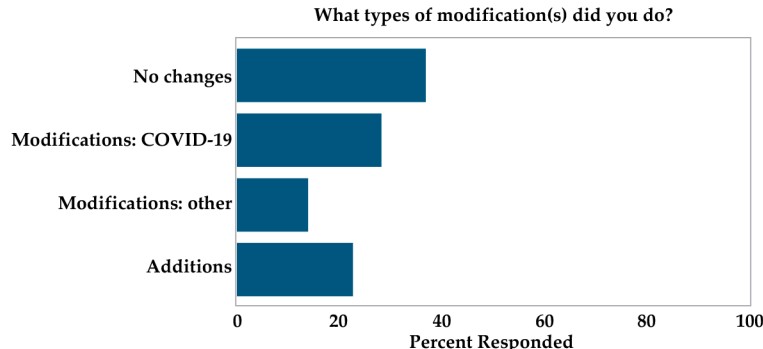

**Figure 12.** EEK! implementation: adaptations made to EEK! for implementation within museum learning contexts during a social disruption, by type.

> "We needed to modify the activities slightly for a virtual format, until we can return to in-person programming. In the scripts for the upcoming videos we plan to use coin flipping for the lab collab game. ...As a take home activity to the Live Program, we will provide the Feel the Beat scavenger hunt. We made these modifications for the live program due to time constraints and wanting to provide an activity that the kids could do with us in the moment".

Other modifications reported focused more on adapting the kits to local needs and target audiences—shortening the time kids exercised before finding their heart rate in one activity, expanding the audience size, or making the directions simpler for younger audiences. As this respondent explained:

> "We modified them to make it easier to lead with large groups in open setting where a participants might not want to stick around. We made smaller sheets with "try this at home" information, or made the overall activity shorter".

To assist in the implementing and adapting of the kits, the EEK! developers provided several planned and ad hoc supports and resources they had identified through their ongoing interactions and co-creation with the participating museums (see Figure 13). As part of the survey, respondents were asked which of the resources they utilized, to rate each resource on a 5-point Likert scale in terms of how helpful it was for program implementation (1 = not at all helpful; 5 = extremely helpful), and to comment on how the resources and training they used helped them to implement EEK!. Nearly all of the respondents (97%) participated in either an in-person or virtual training workshop. Nearly all the training participants rated the training as very or extremely helpful. As one respondent explained:

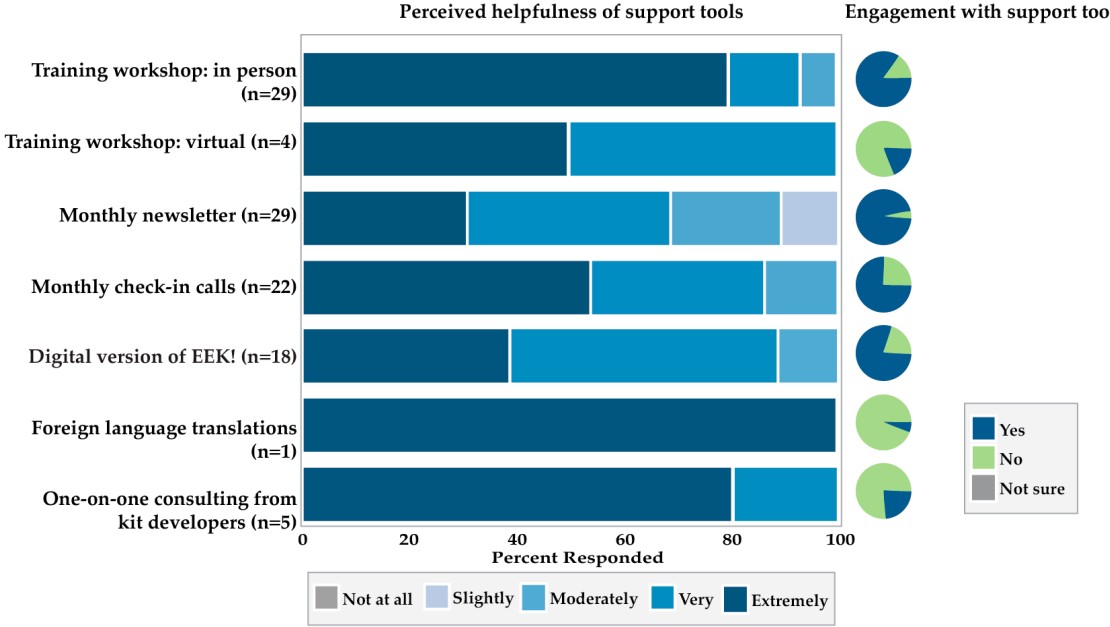

**Figure 13.** Implementation support: use and perceived helpfulness of planned and ad hoc support tools.

> *"The training was a great opportunity for me to try out the activities and work with my peers to consider all of the possible ways we could use them"*.

However, the spread of COVID-19 in the U.S. quickly made it apparent that additional supports would be essential to help the museum partners cope. In response, the EEK! design team instituted the monthly newsletter and check-in calls previously discussed so that the participating museums could stay informed about what was going on with the EEK! program, and obtain support from other museums also grappling with operational changes and staff reductions. As these respondents explained:

> *"The newsletter and check ins have been a really wonderful way to stay in touch with the community and support one another"*.

> *"Seeing how others used the kits validated what we're doing and it was nice to see the slight variations"*.

The monthly check-in calls were also an important collaborative mechanism for discovering participant needs and co-creating supports and solutions with which to address them. For example:

> *"I had mentioned to [EEK! lead] very early on that I need the handouts and especially the permission forms for the research to be translated into Spanish, as that was a significant portion of our demographics that we were reaching out to"*.

A majority (76%) reported participating in these group check-in calls, which they rated as very to extremely helpful on average (mean = 4.41/5.00, SD = 0.72). Nearly all of the

respondents reported using the newsletter (97%), and those that used it tended to rate it as very helpful (mean = 3.90/5.00, SD = 0.96). One of the first supports the museums requested were digital versions of EEK!. Seventy-nine percent of respondents indicated that they used the digital versions of EEK! in their programming and rated them as very to extremely helpful (mean = 4.28/5.00, SD = 0.65).

> *"The digitized activities were a great resource to put on our website to give our audience activities to do while in quarantine. This was very helpful to have for us since we did not have the bandwidth or staff capacity to digitize them ourselves".*

A few participants also requested versions in other languages (Vietnamese, Spanish, Tagalog) of the kit activities so they could be used with local immigrant communities served by those museum partners. While a small minority of the participants (5%) requested and utilized the foreign language translations for the kits, they all found them to be extremely helpful (mean = 5.00/5.00, SD = 0.00). Finally, co-creation was also supported through one-on-one consultations between the EEK! design team and the participating museums. Nearly a quarter of respondents (24%) took advantage of the one-on-one consulting support, and those that did rated it as extremely helpful (mean = 4.80/5.00, SD = 0.40). All in all, the respondents found the resources provided to be very helpful as they grappled with an unprecedented social disruption:

> *"The support offered through the EEK! Project has been over and beyond [what] I ever expected. The resources provided have made my job so much easier. The support through monthly check-ins gave me professional support and assistance in continuing to plan in the COVID pandemic. Having resources available at my fingertips has saved so much time. Knowing I can reach out and receive timely responses is so helpful!".*

*3.4. RQ2c: EEK! Reach to Under-Served Youth*

In total, the survey respondents estimated that they had reached 13,554 youth and their families through EEK! within the first 6 months of the program—either through virtual or in-person programming. However, we also wanted to understand the extent to which the participating organizations were able to engage the targeted audience with EEK!—youth and their families, particularly those from traditionally underrepresented groups in STEM. While it would be disruptive to the intent of the program to collect youth participant demographics directly, we used a number of indicators to measure EEK! reach, including participant organization partnerships (see Figure 14), programming (see Figure 15), and targeted outreach efforts.

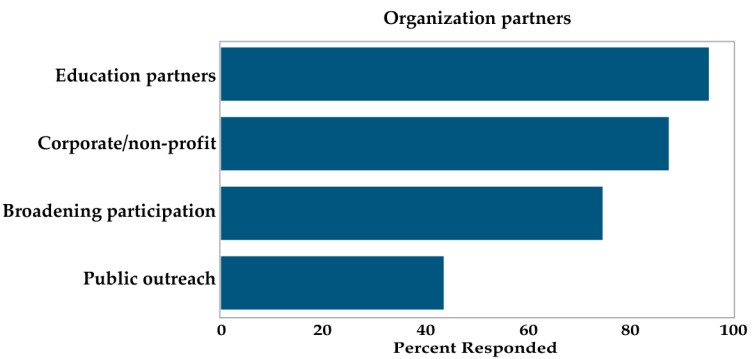

**Figure 14.** Partner types reported by survey respondents.

We considered the types of partnerships the museums had as an important mechanism for understanding the kinds of audiences they were likely to reach. We asked respondents to report on their organizational partnerships, and then grouped them into the following categories: broadening participation partners (organizations serving low-income communities, communities of color, women and girls, people living with disabilities, immigrants, veterans,

or other under-served populations), corporate/non-profit partners (corporations, industry, or non-profit organizations), health-focused public outreach partners (public health, health care, or other outreach organizations), and education partners (preschools, elementary schools, middle schools, high schools, community colleges, universities, organizations providing before/after school programs, camps, and other museums). Similarly, all of the respondents (100%) reported partnering with other education partners, and 78% reported partnering with organizations focused on broadening participation. The vast majority (92%) also reported corporate/non-profit partnerships, and nearly half (46%) reported partnerships with health-focused outreach organizations. These results indicate that the participant organizations were well positioned to deliver EEK! programming to its target audiences.

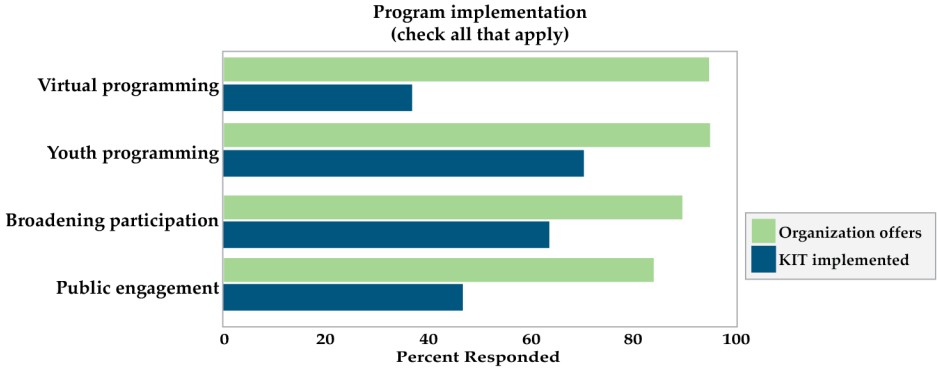

**Figure 15.** Types of programming offered by respondents and EEK! usage within each programming type.

The participating museums were also asked to report whether they offered various types of programming, which were grouped into four main programming types: public engagement (free days, community demos, community outreach, hospital programs, and adult programming), virtual programming (virtual science events, social media posts, web videos, and live streams), youth programming (camps, after school programs, school visits, teacher professional development, birthday parties, family programming, in-person learning labs for virtual school, homeschool programming, scouting programs, and other youth programs), and broadening participation programs (free/discounted admission for low-income visitors, sensory-friendly programming, programming for non-native English speakers, programming for girls, programming for culturally relevant events/holidays, and other programming for under-served populations). The majority of the respondents reported that their organization offered programming targeted at youth (95%) and broadening participation (89%). The majority also offered virtual (95%) and public engagement programs (83%). When asked whether they used the EEK! activities in any of their programming, 70% said they had used it for youth programming, and 63% said they had incorporated it into their broadening participation programming. Additionally, nearly half (47%) reported using EEK! for public engagement programming, and more than a third (37%) used it for virtual programming.

Finally, the participants were asked whether the EEK! activities were used for programs or events specifically targeted at under-served populations (yes/no/unsure), or whether their organization had made any other special efforts to reach under-served audiences with the EEK! program. The respondents that had were asked to describe those programs, events, and other efforts to reach under-served audiences (open ended). Forty-three percent of respondents indicated that they had used EEK! for programs or events specifically targeted at under-served populations, or had made other special efforts to reach under-served audiences with the EEK! program. When asked to describe the programming, events or other efforts targeted at reaching under-served audiences with EEK!, the respondents' comments identified activities that were coded into the following themes: youth focused, broadening participation focused, leveraged partnerships, and response to the COVID-19 pandemic. In addition to events and programs, the comments described additional efforts

to reach under-served audiences that included marketing efforts and leveraging external funding to support those programs and events (see Table 3).

**Table 3.** Programming, events, or other efforts targeted at reaching under-served audiences with EEK!.

| Theme | % of Comments Reflecting Theme | Illustrative Comment |
|---|---|---|
| Youth focused | 100% | *We used an EEK! activity during one of our [State] Engineering Challenge event where 1st-5th graders come to the museum to compete in an engineering competition for a sustained project they have been working on. This program is open to all schools over [State] and is little to no cost with low barrier to entry. Participation in this event is usually a diverse crowd from schools throughout the state.* |
| Broadening participation focused | 85% | *[Broadening Participation Partner Name] is a community partner that [provide[s] students underrepresented in STEM with the opportunities to develop their skills, and explore STEM higher education and career path] Their year-end event [Broadening Participation Partner Name] Week (usually a day-long, in-person [Partner Name] Day event; converted to a week-long virtual event). Part of their programming was hands-on STEM activities. [Museum Name] curated the week's STEM activities and featured Cell Posts as a hands-on, recreate-at-home activity for the day themed "Entrepreneurship".* |
| Partnership based | 85% | *We attended a community fair hosted by the [Corporate/Non-Profit Partner Name] geared for individuals attending title one schools. School participants invited where from the [Community Name] School district and the [Community Name] School district.* |
| COVID-19 response | 23% | *[Museum Name] recently opened up learning pods where we served children on scholarships based on financial need.* |
| Marketing | 23% | *We partner with organizations that serve these populations and market our events to their clients.* |
| Leverage external funding | 15% | *We are part of the [Education Partners Initiative Name] initiative and we routinely fundraise for our [Broadening Participation Program Name] that gives free admission to our museum and summer camps.* |

Combining those who used EEK! with programming, events, or other efforts targeting under-served audiences with those who reported implementing EEK! with their broadening participation programming, 73% of the museum respondents explicitly reported using EEK! to reach under-served youth and broaden participation. Considering that 78% of the museum respondents reported implementing EEK! at some point after the program launch, that leaves just 5% who did not explicitly report using EEK! to broaden participation. Nevertheless, these organizations may have been able to reach under-served youth through their less-targeted EEK! implementation activities. We also note that 87% of the respondents who did not report using EEK! to broaden participation did report that they either offered broadening participation programming or had broadening participations partners, indicating that they may have the capacity to broaden participation using EEK! in the future. In total, 94% of the museum respondents reported that they had either used EEK! to broaden participation or had the capacity to do so through their existing broadening participation programs or partnerships. Nearly all of the respondents indicated they intended to use the EEK! kits again in the future, with 94% saying they definitely or probably would. As the U.S. education ecosystem continues to grapple with ongoing social disruptions, it will be important to continue to support informal learning organizations over the long term.

## 4. Discussion and Implications

Though this paper sought to explore the utility of higher education-led, co-created engineering education interventions (EEK!) to the museum community, we were also able to gather important information about the impact of a societal disruption (COVID-19) on

a critical component of the education ecosystem, museums. Five key recommendations emerge via this study. Recommendations one and two are offered as a result of the successes of EEK!. Recommendation three is a lesson learned and an opportunity for both EEK! creators and the broader community to strengthen the efficacy and impact of future programs like EEK!. Finally, recommendation four and five are lessons learned to build better resilience in the face of social disruption.

### 4.1. Recommendation One: Co-Create Resources with Your Intended Users as True Partners to Facilitate Adaptability

This study reveals several interesting findings pertaining to disseminating and supporting engineering education programs in museums during times of social disruption. First, the data reveal key information about the respondents; they experienced the ongoing impacts of COVID-19, with the majority having to shift to virtual engagement and/or operating at limited capacity with reduced staff sizes. Despite these challenges, they still implemented and adapted the EEK!, with nearly all respondents expressing a desire to continue using it.

This study draws attention to how robust, thoughtful co-creation can result in resilient resources and partnerships that build resilience and adaptive capacity during times of disruption. The co-creation involved in the design and implementation of EEK! directly resulted in key resources that allowed museums to adapt on the fly, such as a monthly newsletter, digital versions, and community check-ins. Without the collaborative efforts, those key resources likely would not have emerged or been as successful at meeting the intended users' needs. Through thoughtful relationship building in the service of co-creation, institutions like universities and K-12 schools can partner with museums, particularly science museums, to strengthen their efficacy and engage diverse audiences. Indeed, despite the massive impacts of COVID-19 on our partner institutions, the vast majority (78%) were able to implement EEK!. All of those who did not implement it, attributed that directly to the logistical challenges caused by their steep reductions in resources and staff. Many of the benefits of the strengthened partnerships established through EEK! continue to hold value beyond the duration of the global pandemic and the social distancing it required. For example, science museums and university outreach programs have often wrestled with the logistics of reaching rural communities and other low-income populations with limited access to transportation. COVID-19 necessitated rapid innovation to create distance and virtual delivery tools which could be used with historically hard-to-reach communities. More importantly, it forced society to acknowledge and begin work to address the uneven distribution of resources, such as functioning computers and stable internet connections, that has always played a role in access to educational opportunities.

As a final note, we use the term "true partners" to emphasize the importance of sharing power with those you partner with. This may involve waiting to define a project until ample input has been provided, shifting a project in the face of feedback, or advocating for your museum partners' perspective in meetings they may not be present at. A more extractive approach, where museums are used solely to provide input and refine a product, may not yield the same outcomes.

### 4.2. Recommendation Two: Prioritize Building Connection and Community

One of the most valuable insights from this study may be tied to the types of interventions that were most useful for participants. Of the three COVID-19 support tools that were offered, the monthly newsletter was the most broadly used; however, the community check-ins were rated the most useful, suggesting that the ability to collaboratively troubleshoot and learn from peers during community check-ins is a helpful resource with which to support program implementation and adaptation. In our observations, the deep partnerships that co-creation provides are helpful for quickly identifying opportunities in general, and more so in fast-moving and uncertain times.

Furthermore, the groundwork was laid for this through our training model that focused on community discussion. Though it would have been more cost effective and expedient to conduct the initial training for the kit recipients virtually or as a self-paced course, we opted for real-time, brainstorming events with one of the goals being to build community and familiarity. We believe it was this familiarity that allowed for the fruitful dialogue and resource exchanges that regularly occurred that both supported the participants and identified opportunities for future informal engineering engagement opportunities. For example, during a community check-in, science museum educators requested gamification of EEK! activities for online usage. Such a resource would provide deeper engagement for youth learners and free science museum educators to focus on contextualizing the learning experience. It would allow universities to reach beyond their science museum partners and use the games for their own outreach programs. The creation of the game could even potentially allow for a venue for undergraduate and graduate student learning if the creation of the activities was facilitated through a computer science or engineering course. Finally, since interactive, multimedia learning environments are currently heavily studied, this could provide a research opportunity for faculty and students.

### 4.3. Recommendation Three: Identify Opportunities for Tighter Programmatic Coupling

Though more than three-quarters of the participants reported having implemented EEK! (78%), and even more expressed a desire to use EEK! in the future (94%), more than half did not report immediate plans to do so (62%). They expressed that this was primarily a result of needing to focus on their key programming. As previously noted, we believe this may be indicative of two key challenges that they are facing: resource scarcity and fatigue. It is notable that during this time of social disruption, programs that focus on broadening participation within under-resourced communities may have been deprioritized due to a need to focus on finding ways to stay in operation, or the need to adapt to new social restrictions. With most organizations demonstrating the ability to adapt, one may postulate that finding ways to maintain these adaptations may be particularly challenging as the lingering impacts of the pandemic continued, despite museums' impressive ability to quickly adapt initially. As concerns about burnout rise among businesses and universities [77,78], it may be that museums were not exempt from this COVID-19 consequence either.

Indeed, when looking at the perceived fit of EEK!, programming fit was assessed the worst as compared to the fit with budget and mission. This challenge needs to be explored more thoroughly because the key components of EEK! (engineering and the heart) do generally align with the programming components of most science museums and, in particular, ones where STEM is a priority. Others looking to design similar interventions should consider conducting programming audits for the intended users prior to the early stages of the design process to ensure tighter coupling between educational interventions and current programming.

### 4.4. Recommendation Four: Support Museums to Support Schools

Museums often partner with schools to offer enrichment activities for students (field-trips) and teachers (professional development). Nearly all of our respondents (97.3%) reported partnering with schools, and identified activities ranging from virtual visits to creating curriculum as part of their partnership activities. Through partnering with museums, higher education can more effectively and efficiently support schools through relying on museums' expertise and pre-existing relationships with formal learning institutions. The benefits of co-creation, in this context, are the accommodation and understanding of educator experience and confidence in teaching and learning engineering and, on the university's part, the support for the adaptability with which to effectively train educators in pedagogical and content gaps in engineering contexts.

There are many potential opportunities for higher education to partner with science museums based on the reported innovations of the respondents. Science museums

have partnered with universities to provide remote activities to lessen childcare burdens, have partnered with schools to provide supplemental curriculum, and have partnered directly with parents to create daycare/learning pods for those in careers that do not allow them to work from home. Given that the pandemic has exposed disparities within the workforce, including academia, that affect women and those holding marginalized racial identities [79–82], institutions of higher education have the opportunity to learn from these programs and extend partnerships for the long-term that will sustain these resources for working parents and their children.

*4.5. Recommendation Five: Provide Advocacy and Support for Museums and Informal Learning Institutions*

Though informal learning environments have the advantage of not being tied to particular learning standards or disciplinary requirements, they also rely more heavily on revenue to cover operation costs. Indeed, AAM [76] reported that, in the first six months of the pandemic, the responding museums each lost an average of $805 K in revenue, and 29% reported they were at risk of closing as a result. This aligns with the Association of Science and Technology Centers' estimate of a loss of $600 million across their 500-institution membership body [83]. This means that there is a significant danger of being deprioritized if they do not shift and demonstrate that they are highly valuable and engaging to teachers and parents. Many museums seized new opportunities to create access to resources in spite of the many barriers introduced by COVID-19. For instance, some participants offered pay-per-play digital escape rooms that included trivia clues or other learning opportunities embedded within the activity. School field trips were adapted into virtual excursions, with live workshops and video tours of museums around the world, including those that may not have been locally accessible to students. Those lacking virtual access could receive mailed activity kits they could use directly in their home, similar to those produced by CELL-MET.

In acknowledging the role that museums play as centers of culture and science communication, we must also acknowledge that society must work to help preserve them during this challenging time. The first step in this process is acknowledging and formalizing their role as critical to the education ecosystem, rather than presenting it as optional. The establishment of sustainable, ongoing partnerships with learning institutions is a way that higher education can support these cultural institutions. Further, the development of such partnerships are relatively facile. Museums demonstrating an ability to adapt and implement new activities and programming in the face of the staffing and fiscal burdens created by a global pandemic are a testament to their strength and flexibility. Their societal benefit is magnified by their willingness to partner with other institutions, elevating partners through their innovation. Universities can contribute by advocating for funding, supporting them in grant proposals, or utilizing their higher education fundraising skills to ensure that financial constraints do not prevent informal learning institutions from making as big of an impact as they are capable of. This work is not without benefit to higher education. For the teaching and learning of engineering, museums and their staff can enhance universities' impact in three key ways. First, museums have a reach that far surpasses that of most university-based engineering outreach programs, even small centers can reach tens of thousands of students a year. Second, as expert informal educators, they are adept at building excitement and curiosity and getting young learners and their families excited about engineering. Finally, the resources they create can be used in the engineering outreach programs offered by universities directly, so those offerings can also be improved.

**5. Conclusions**

The need for formal–informal education relationships arose when the world suddenly required adaptability in the face of COVID-19 closures. Schools experienced great disruption, but through an education ecosystems approach—where when one area is lacking, another part of the education ecosystem can scaffold learning experiences and opportunities for teachers and students—museums were poised to help address this gap. By all

accounts, museums were greatly impacted across all levels of operation: from staffing reductions and shifts, to lost or limited use of physical space, and the transition to virtual or self-guided engagement. They also lost critical partnerships through the loss of school and family visits. This is particularly challenging given the highly physical nature of informal learning environments, and the reliance of many museums on payments associated with in-person visitors, such as families, school field trips, and event programming such as summer camps. Despite these challenges, museums demonstrated great resilience and innovation by swiftly adapting, and in some cases expanding, their engineering activities, while also maintaining their core programming by building distance learning opportunities and finding ways to rethink their exhibits to connect people to their physical spaces despite their own inability to be in the museums. Thoughtful university–museum partnerships can be an effective way to support STEM education for K-12 learners and, more specifically, address systemic inequity and fill learning gaps that result from social disruption.

*Looking Forward: Applicability to Other Contexts*

Formal learning institutions face a different suite of problems compared to informal learning institutions. They share the challenge of balancing staff and student well-being with the need to promote learning progress, meet educational standards, and maintain access for all students. Disruptions in the spring semester led to intentional planning and modified learning in the fall, the introduction of technological tools, and pedagogical shifts to account for barriers introduced by the pandemic. However, even though meeting learning standards at any level—whether K-12 or higher education—continues to be challenging, the demand for formal learning, particularly in the K-12 environment, remains constant because of the state and federal requirements associated with it. As such, CELL-MET has created programs to support K-12 teachers in adapting the kit for use within formal learning environments. Specifically, EEK! offers a yearlong community-focused program for a small cohort of teachers to implement programming, adapted to fit within their unique classroom setting. Additionally, we are offering expanded professional development opportunities for informal education personnel to continue to strengthen their capacity to expand the experiences K-12 learners have with EEK!, CELL-MET research, and the field of engineering as a whole.

Museums' flexibility and ability to adapt to a diversity of audiences and ages makes the materials and the models they employ beneficial beyond their immediate settings. Furthermore, their practice of embracing the "whole learner" [84] is a concept that has been demonstrated to have a positive impact on learners from diverse backgrounds and undergraduate student success. Continued partnerships with museums and their models of engagement have strengthened the impact of our outreach and professional development efforts in the COVID-19 educational setting, while also demonstrating the value of intentional, ongoing partnerships moving forward.

**Author Contributions:** Conceptualization, S.L.R.; methodology, L.C.M., formal analysis, L.C.M. and S.E.S.; data curation, L.C.M.; writing—original draft preparation, S.L.R. and S.E.S., writing—review and editing, S.L.R., L.C.M., M.D.H., S.C.H. and A.J.M.; visualization, A.J.M. and S.L.R.; supervision, S.C.H.; funding acquisition, S.C.H. All authors have read and agreed to the published version of the manuscript.

**Funding:** This material is based upon work supported by the National Science Foundation under Grant EEC-1647837.

**Institutional Review Board Statement:** This study was conducted in accordance with the Declaration of Helsinki, and the protocol was approved by the Internal Review Board of North Carolina State University (23624).

**Informed Consent Statement:** Informed consent was given by all the subjects included in this study.

**Data Availability Statement:** The data presented in this study are available on request from the corresponding author.

**Acknowledgments:** We would like to acknowledge and thank the many educators who have used and adapted EEK! We would also like to extend a special thank you to the many people who heavily contributed to the creation of EEK!: David Bishop, Brenda Boddiger, Edgar Cardenas, Chris Chen, Sarah Craven, Claire Doddman, Kevin Farmer, Stacey Freeman, Rachael Jayne, Joerg Lahan, David Laubenthal, Rebeccah Luu, Leah Melber, Ana Montesdeoca, Bekka Nolan, Leonardo Svarc, and Alice White. An additional thank you to Rockman et al. for their evaluation support.

**Conflicts of Interest:** The authors declare no conflicts of interest.

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
