# Peer review of "University–Museum Partnerships for K-12 Engineering Learning: Understanding the Utility of a Community Co-Created Informal Education Program in a Time of Social Disruption"

_education, doi:10.3390/educsci14020146_

Round 1
Reviewer 1 Report
Comments and Suggestions for Authors
Reviewer 2 Report
Comments and Suggestions for Authors
General Comments:
The paper starts by posing two questions:
1) How were science museums impacted by and responded to COVID-19
2) How can co-creation help universities partner with science museums for mutual benefit, particularly in the face of complex challenges?
The first question is well addressed by the paper (specific comments below) but the second one is not. By the end of the manuscript I was left curious about the structure and components of the KIT, unclear of its educational intent, not able to clearly see the mutual benefit to universities and museums (informal learning institutions would be better), and unsatisfied with the level of detail about the co-creation and adaptation components of the product development process (the survey data could reveal more). Readers should be able to learn from this experience about how to co-create more meaningfully. Also, is there anything in the survey about the reaction from the end users? That is a black box and needs addressing in some way.
The manuscript would benefit if it spent more time on the product itself and highlighted the learning objectives, how the institutions did in fact collaborate (get into some specificity of the adaptations – it is currently very high level), and responded to the feedback (Figure 2 is insufficient) – were any of the user institutions surveyed involved in the development of the KIT?
From what I gleaned from this manuscript, this product was not co-created with the end users/the communities targeted until the implementation phase and the few of those informal institutions that may have had input on the product may or may not have used the product (that was not made explicit). We learn a lot about the challenges of getting the KIT to market during COVID but that is not helpful for readers of this journal who may be developed products of their own and want to better plan co-development with users who need better STEM learning opportunities. There may be two papers here instead of one given the information presented. One could focus on the COVID challenge and the role informal institutions can play given the KIT experience and the other is the KIT development, implementation, and evaluation that was impacted by COVID. It could stay as one paper but I think the second question should then be the main focus and the first question becomes proof for why a collaborative approach creates flexibility for implementation when there are complicating external factors at play.
Specific comments:
Line 10 – what is meant by pressure test in this context?
Line 17 – Pick K-12 (capital K) or k-12 and be consistent throughout
Line 79-80 – the partnerships are not made clear in the write-up
Line 154 – boundary objects – what are they in the KIT context?
Line 208 and 209: what ideas are being suggested? The boundary objects? There is a lack of detail (lots of generalities)
Line 214: Here is one of the few places in the article any content of the purpose of the KIT is mentioned and it begs for more description of the KIT – especially given the target audience (another is on line 546 where there is discussion of the heart rate being taken … the mystery of the content of the KIT makes it hard to put the description into a useful context.
Line 234: What are the topics?
Line 240: This description of hands-on, mind-on is intriguing – how does the KIT product incorporate this? A visual or some elaboration of the KIT would be helpful here. Also, how is the KIT flexible?
Lines 257-259 and Figure 1: Surprised if trying to do participatory research and engagement that the end users (underrepresented students and perhaps teachers) were not part of the design process – how does the adaptation stage work? It feeds back to dissemination (not design) so what was changed? What are capacity-focused trainings and how did they feed into the implementation? Collaboration and resource sharing seems to come through the ongoing support – not seeing how these two things are distinct. Were the museums involved in the development among the 51 informal learning organizations that implemented KIT (and were surveyed/evaluated)? If yes, their answers themselves may be worth isolating in the analysis.
Line 391/Figure 2 – here we see an adaptation that was made in the product development process but it is a shallow example. Identifying activities participants enjoy is helping them how? Are the medical benefits identified that relate back to learning objectives? Were any of the target audience included in the feedback (especially students and teachers from underrepresented groups?) Several of the activities listed have a cultural component – listening to music isn’t included or dancing or the arts – why is there no space for students to add their own activity?
Line 487 (Figure 5) – B . what does all time versus current mean? Current refers to EEK! Implementation currently? (The menu for A is clearer)
Line 499: Accommodations and Adaptations
Examples here would prove very helpful to readers. The adaptations are organized into 1) operational 2) theoretical and 3) additions – what is the theoretical basis of the KIT? Operational (delivery?). If I missed it, it needs to be highlighted more because this comes into play here. The application of qualitative content analysis needs to be elaborated – what criteria was used for each? Could you discern more about the type of users based on the needed changes? The adaptations of making them more experiential so they fit with existing programming is interesting – more information about that would be of use especially for school integration. Overall, this section needs more detail because the KIT theory, components, and activities remain a mystery.
Line 511: how are training workshops considered co-created or collaborative experiences? It sounds like the developers made themselves available during COVID and supported their end users but a connection is not clear on how these were co-created.
543-544 – how are shortened exercise times, expanding the audience size or simplifying instructions operational? Why were these changes needed (it would be interesting to see which locations were making these types of changes and why).
Line 554: Figure 6
Here we learn that 20% of the institutions serve (or try to target) populations underrepresented in STEM – can you isolate their answers to learn more about what they had to do and why to better integrate/use the KITs? In the theory section a lot of time is spent on how important reaching underserved populations are but we are not unpacking the survey to help us see how to do it better or what needs to happen (especially if representatives of those populations were not specifically targeted up front in the collaborative design phase).
Line 584: How were intended users equal partners? Maybe they were but not clear in the paper.
Line 616: This is perhaps the most useful aspect of this paper – the ways the developers stayed in touch with the users during implementation seems a sure way to see how the users are able to implement the product and offers guidance (hopefully for efficacy) so the learning objectives can be maintained even as users adapt to their needs. It is not clear how this was co-creation so much as building connections – if the authors clarify what they mean by co-creation and collaboration in the context of the KIT that would be helpful for readers.
Line 628-629 – here we learn a lot more about the content of the ongoing support and collaboration and resource sharing and capacity-focused trainings … the real time brainstorming supports the idea of a partnership – and the gaming component has real potential so long as the end users are involved from early on. The community check-in component of this paper gets lost until the end and would be helpful if it was more developed early on in the paper.
657: The programmatic fit with institutions is where earlier partnering/collaboration could (perhaps) better serve all involved. Bringing in schools and museums together with the design team would help identify programmatic gaps (especially in terms of subject matter teachers have to address for testing).
Line 721: Conclusion – here you seem to make the case for two different papers. One is that COVID-19 and its impacts on implementation highlights the critical role museums and other ILI (informal learning institutions) can play for educating the public and you also have a paper to talk about how you are developing and implementing a product that had difficulty during COVID BUT COVID provided an opportunity for building better collaboration with those implementing the product.
758-59: What are the manipulable materials? And the culture box that is introduced in the second to last paragraph is tossed in – why bring this in at the end? The conclusion does not tightly bring together the ideas presented in the paper.
Grammar Issues noted:
Line 6: An activity kit (KIT)
Line 8-9: how were science museums were impacted by and responded to COVID-19 and (b) how can co-creation co-created products can ?? help universities partner with science museums for mutual benefit, particularly in the face of complex challenges?
Line 9: Through this Under these circumstances, the authors pressure tested their model of product development and implementation
Line 12: 97.3% of what closed? Museums? (given the informal learning institutions not all being museums, some specificity is needed here – from I read this is the percent of those informal learning institutions surveyed)
Line 17: Inconsistent (k-12 versus K-12)
Line 393: The word Materials is missing the M
Reviewer 3 Report
Comments and Suggestions for Authors
I appreciate the perspective of this article, as museums and other informal learning environments are indeed a vital, and undervalued, part of the learning ecosystem. However, at several points, this manuscript read more as advocacy for museums than as a report on research, particularly research on engineering education, the topic of this special issue. Overall, I don't feel like I learned much about engineering teaching and learning. Perhaps this manuscript would be a better fit for a journal on educational change or museum studies.
Additional pieces that I think are necessary to improve the quality of the manuscript include:
In the Introduction:
Greater engagement with relevant literature and theory. For example, the authors invoked the idea of a learning ecosystem, but didn't define what they meant by this or draw on the relevant work of scholars like Uri Bronfenbrenner (ecological systems theory) or Nichole Pinkard (healthy learning ecosystems incorporating informal and formal institutions).
Similarly, the authors discussed the distinction between informal and formal learning, but with very sparse citations. The authors might consider reading and citing the following chapter, which distinguishes informal and formal learning in ways well aligned with but a bit more nuanced than what it currently described in the article:
Bransford, J., Vye, N., Stevens, R., Kuhl, P., Schwartz, D., Bell, P.,... & Roschelle, J. (2006). Learning theories and education: Toward a decade of synergy. Handbook of educational psychology.
Also, in the introduction, the authors might address the drawbacks of informal, museum-based learning, like lack of access due to cost and transportation. They should also discuss the limitations of museums to support pandemic-era learning. For example, it is not clear to me that museums have greater expertise than schools at facilitating remote learning, and a stronger case could have been made for their ability to engage learner interests. I was also not convinced by the claim "Indeed, it is possible that families are relying more on informal education organizations as a result of pandemic-related disruptions to formal education organizations." in lines 330-331. This wasn't supported by evidence, and it wasn't clear why museums specifically would be more relied upon during the pandemic than schools, since they are still indoors and subject to many of the same health and safety concerns as schools. And while I appreciate the argument that less attention was paid to the impact of the pandemic on museums than on schools, it is important to understand and acknowledge that this is in large part because, in our society, schools function not just as educational institutions but as child care and as sites for primary social interaction with peers, in a way that museums do not.
Finally, the authors discuss a tension in the literature on dissemination of innovations between a desire for fidelity of implementation and a need for adaptation. However, they do not engage deeply with this literature, including prior approaches aimed at balancing fidelity and adaptability, not do they really explain how or whether they believe their program might strike a balance between these two goals.
In the Materials and Methods section:
First, there is a typo in "aterials and Methods". Second, I would have liked to have heard a lot more about the kits themselves. How are the activities structured? What types of materials are provided? What specific set of topics are covered? What type of feedback to learners receive? Figure 2 was somewhat helpful here, but the depicted "activity" wasn't really an activity, it was just a question presented in different ways. What did the rest of this activity look like?
It would also have been helpful to know exactly when the survey was completing, to provide context for the data on museums adaptations to Covid conditions, which obviously changed throughout the pandemic (i.e., many more institutions were closed in spring and summer 2020 than in Fall 2020 or throughout 2021).
I would also have liked to see more detail on the format and content of the qualitative questions asked and how responses were coded.
In the Results section:
First, it looks like the authors forgot to blind the program name in all the figures.
Second, reporting of results seem to focus mostly on Covid adaptations, rather than on engineering teaching and learning.
Third, there were some claims in the results that we not adequately substantiated with data. For example, in line 466-467, the authors wrote, "This resulted in 22% of the organizations reporting never implementing the kits, most as a result of issues related to staffing such as the intended facilitator being laid off or furloughed." It is unclear how the authors new the cause of failure to implement. Did this come from survey responses? If so, it would be helpful to share specific quotes from the surveys that substantiate this claim.
In the discussion and implications:
Again, most of the key takeaways here appear to focus on the adaptation of educational innovations rather than on engineering teaching and learning specifically.
Many of the conclusions that are drawn are also not new to the field of educational change research (e.g., co-creating resources with partners). This is where a deeper engagement with prior literature would be helpful. The authors might consider reviewing articles in the Journal of Educational Change or scholarship on research-practice partnerships and community-engaged research/design.
There was also a return in the discussion to a tone of advocacy for museums rather than simply discussing direct implication of the study results.
Comments on the Quality of English LanguageThe quality of English language was overall fine. There were just a few small typos.
For example, in the intro, line 28 "tthat"
In the Materials and Methods section header, the M in Materials is missing.
In the Discussion and Implications section, lines 581-582 “Finally, recommendation four and five are lessons learned, which education partners can use to” The rest of this sentence appears to be missing.
Round 2
Reviewer 2 Report
Comments and Suggestions for Authors
Thank you for addressing my big issues with the development process and how you were targeting underrepresented groups in STEM. This version is much better along those lines. I especially appreciate figures 2, 3, and 6, as well as Table 1 and especially Table 3. Really like the way you brought in how KIT can be reaching the audiences of greatest interest. While the paper has come a long way, there are still some general issues that need to be addressed.
First, there is a tension throughout the paper between using institutions of informal learning and science museums and museums (the main questions of the paper use “science museums” in question 1 and “museums” generally in question 2). The paper needs a careful edit around which terms are used when and why. Break down the numbers up front (pull in your info from lines 709-711) OR just say you focused on museums, mostly science but also children, history, and natural history museums OR were there other types of informal institutions than museums that informed the KIT/that the KIT was sent to? From my read of the paper, I would stick with museums as there are many different types of informal institutions of learning beyond museums (e.g., botanical gardens, nature centers, field stations, arboretums).
Why in the title and in question 2 is the focus on university-science museum partnerships? Is the engineering research center part of a university? If yes, be explicit else it appears more accurate to refer to the project as a research-museum partnership or product development partnership or something else. This needs to be addressed throughout the paper.
I think there is a distinction to be made in the paper between co-creation and collaboration as well as the partners who helped with development of the product and those who gave feedback during implementation – were any of them the same? Is there much overlap? I am still not clear on that point.
The collaborative chain-link model of innovation makes a great deal of sense given your process of product creation and implementation, and there seem to be varying levels of co-creation that happened at each stage of Fig. 1. However, I do not see you (the museums and research center/product developers) as equal partners (equal stakeholders) in this project (see recommendation one, line 1092). You are a funded research center designing something that has a STEM education purpose museums can use (and often need). There is feedback given from the museums and users/families and adjustments made to the product and its implementation so you could say this is a product made collaboratively and has aspects of co-creation and even participatory research as you indicate, but the main stakeholder is clearly the research institute that conceived of the product and its core learning objectives.
Finally, I found myself trying to edit quite a bit – the writing has redundancies. I suggest a final editing pass to tighten things up.
Specific comments:
Lines 8-9: referring to schools as promoting “standardized cognitive learning” is odd – they promote standardized teaching which you explain in 1.1.2
Lines 13-15: The main questions are written differently here than they are later in the paper (lines 651-654) … the first question should be slightly rewritten for grammar. Also consider how you are going to capture the museums (science, children, history, and natural history museums? OR science-oriented museums or ?)
1) how were science museums impacted by and how did they respond to COVID-19 (note the slightly modified sentence)
2) to what extent were museums able to implement and adapt KIT to reach underserved youth in the face of social disruption?
Line 18: awkward/redundant first sentence. Suggest a rewrite. Something like:
When the world was experiencing social disruption from the spread of COVID-19, the authors realized they had an opportunity to test the utility and adaptability of their model for co-creating engineering activities.
Need a better transition to the survey … suggest starting with:
Approximately six months into the launch of KIT and the global pandemic, a 29-item survey was distributed to KIT-recipient institutions to determine how well the program was working given the extensive social disruption wrought by COVID-19.
Lines 23-26: Note how you refer to and transition between museums v informal institutions of learning here. Also, this seems like a run on sentences and is missing parallel structure (specifically, “…with 70% of institutions creating new virtual programming”).
Line 185: Here you seem to be implying that exhibits, activities, and demonstrations are all boundary objects – is that true or are these events providing boundary objects (such as a piece of art) for the public to engage with? The writing needs some cleaning up.
Lines 196-200: The idea is a bit muddy – suggest going to the heart of what you are trying to say: “Though the literature on boundary objects focuses on benefits to the group, Perhaps the most important benefit of boundary objects is for novices to gain agency through the process …” Then talk about KIT and how it brings in the boundary objects in a new sentence.
Line 350-351- university-museum partnerships (research center-museum partnerships?)
Line 373: Instead of adapted, you are better off describing this as “… programs that were not adaptable.” The essence in the power of your model is that you are able to make adaptations – not adapt to one thing. Subtle difference but seems central.
In general this section needs a bit more editing/paring down.
Line 412-414: Were any of these experts in working with underrepresented youth?
Lines 417-420: Maybe I missed this in all the edits, but were these testers also some of the same end users and were they representative in some way of underrepresented groups? Struggling with fig. 1 – there were 69 informal learning experts? And from 16 institutions (museums?) you reached 14,000? I understand teachers were not an intended audience directly. It is an impressive amount of input and I don’t think the figure captures it well.
451-453 – Aren’t these objects boundary objects? Seems a place to bring in the theory.
Lines 568 – 574: Really interesting to learn about how museums stepped up during the pandemic
Lines 651-654: The research questions here differ slightly from the questions posed at the start of the paper (mentioned above)
Line 693: Table 1 – will you share the complete survey as an appendix? I think it could be helpful for other research groups
Line 803: In Table 2 I would remove No Comment - 0%
Line 986: …beyond WHAT I ever expected (this is in a quote so it may not have been transcribed correctly – or you may need to add it for easier reading)
Line 992: this section (3.2) really adds a lot to the paper!!
Analysis comment: The 14000 reached from figure 1 is a number based on the surveys? I ask because you could bring in some of those numbers for readers to get some sense of how many could potentially be reached in section 3.3.
Line 1081: here higher education is mentioned – please clarify research center v higher education/universities
Lines 1085 – 1090 could be incorporated directly into each recommendation section.
Line 1092: “as equal partners” is problematic - while there seems to be clear moments of co-creation during implementation, it is not clear that you were equal partners - the decision-making power rests with the research center. (See my general notes)
Line 1185: university v research center
Comments on the Quality of English LanguageSee comments to the authors above.
Reviewer 3 Report
Comments and Suggestions for Authors
The manuscript is greatly improved on all fronts. I really appreciate the authors thoughtful engagement with comments and corresponding edits including deepening their engagement with prior literature, clarifying their explanation of their design and research methods, increasing the transparency of their analysis and evidence for claims presented in findings and discussion and clarifying their contribution to the field of engineering education. I also found the additional tables and figures to be particularly helpful.
I believe this manuscript is now suitable for publication. However, I would suggest a final proofread for typos and grammatical errors. For example, in the section on ecological systems theory, 'chonosystem' should be "chronosystem"
Comments on the Quality of English Language
I would suggest a final proofread before publication for typos and grammatical errors. However, overall the quality of the English language is fine.
Author Response
Dear Reviewer 3,
Thank you very much for all of your advice and positive feedback. Per your advice, we have made several more passes to further catch typos and further refine for clarity.
We truly have appreciated this peer review process and the resulting paper.
Thank you.